# Social-R1: Enhancing Social Intelligence in LLMs through Human-like Reinforced Reasoning

## Abstract

Recent advances in reinforcement learning with verifiable rewards (RLVF) have elicited strong reasoning abilities in large language models (LLMs) on objective tasks such as math and coding, yet social intelligence—the capacity to perceive social cues, infer others' mental states, and interact effectively—remains underexplored. We argue that progress has been hindered by the simplicity and homogeneity of existing social datasets, which incentivize shortcut solutions over genuine Theory-of-Mind (ToM) reasoning. To address this, we introduce **ToMBench-Hard**, a challenging, multi-dimensional multiple-choice benchmark that rigorously evaluates ToM (e.g., perspective-taking, belief revision, and deception), exposes limitations of current LLMs, and provides verifiable outcomes for reinforcement learning. Training with RLVF on ToMBench-Hard using only outcome-based rewards already yields clear improvements. Motivated by the role of human-like mental processes in social cognition, we further collect diverse reasoning trajectories and train a social thinking reward model that scores trajectory quality—rewarding accurate perception of social cues and ToM-consistent inference prior to answer generation. We combine these signals in **Social-R1**, a reinforcement learning framework for social reasoning that integrates outcome and trajectory-level rewards. Across SocialIQA, SimpleToM, EmoBench, and MotiveBench, Social-R1 consistently outperforms strong reasoning LLMs; notably, Social-R1-4B surpasses LLaMA3-70B on all benchmarks despite the latter having more than ten times as many parameters. These results show that outcome-based RLVF substantially improves LLMs' social reasoning while process-level thinking rewards provide additional gains, underscoring the importance of supervising the reasoning trajectory to foster human-like social intelligence in language models.

## 1 Introduction

Social intelligence is a critical capability for large language models, underpinning their effectiveness in domains such as emotional companions, personalized education, healthcare, and cross-cultural communication(Sorin et al., 2024; Pinto-Bernal et al., 2025; Vijjini et al., 2024). In humans, a core foundation of social intelligence is social reasoning-the ability to infer others' mental states and interpret them within social and contextual norms(Weis and Süß, 2005). This ability plays a central role in decision making (Sabbagh et al., 2006), social cooperation (Warneken and Tomasello, 2006), and empathetic communication (Lamm et al., 2007). Consequently, cultivating genuine social reasoning is essential if large language models are to achieve higher levels of social intelligence. To this end, researchers have explored diverse approaches to enhance models' social intelligence, including prompt-based approaches (Jung et al., 2024; Gu et al., 2024), model-based approaches such as Bayesian inference (Zhi-Xuan et al., 2022; Jain and Krishnamurthy, 2024), supervised fine-tuning (Wu et al., 2024; Tang and Belle, 2024), and post-training reinforcement learning (Hou et al., 2025; Lu et al., 2025a).

| Ability | Belief: Location False Beliefs |
|---------|-------------------------------|
| Story | Mary and Mike are wandering around the factory. They see a **transparent** box and some cabinets, and they find a hammer on the box. Mike then leaves the factory, and Mary moves the hammer into the box. |
| Question | When Mike returns, where does Mary think he believes the hammer is? |
| | A. On the box  B. In the box  C. In the cabinet  D. In the cupboard |
| Answer | B. In the box |

**Thinking** Initially, Mike and Mary both see the hammer on the transparent box while they are together in the factory. Mike then leaves the factory, and while he is gone, Mary moves the hammer from the top of the box into the box. By the time Mike returns, he would not have seen Mary moving the hammer into the box, as he was outside when she did it. Therefore, Mary would likely think that Mike believes the hammer is still where he last saw it, which is on the box, because he did not witness the change of location. Mary thinks Mike believes the hammer is still On the box. <Answer> A</Answer>

**Thinking** Both Mary and Mike see the hammer resting on the transparent box. Mike then leaves; while he is gone Mary puts the hammer inside the transparent box. The box is **transparent**, so its **contents are visible** from the outside. When Mike returns, the first thing he can do is look at the transparent box; he will immediately see the hammer inside it. Therefore, upon returning, Mike's belief about the hammer's location will be " in the box." Hence option B is the correct answer. <Answer> B</Answer>

Figure 1: A sample from **ToMBench-Hard**, where arriving at the correct answer requires genuine social reasoning (Right) rather than relying on surface-level shortcuts (Left).

However, despite growing research in this direction, fundamental limitations remain. First, existing social benchmarks are overly simplistic and fail to evaluate whether large language models truly engage in social reasoning (Lu et al., 2025a; Mousavi et al., 2025). Many contain exploitable patterns, enabling models to obtain high scores by relying on superficial shortcuts rather than authentic human-like reasoning (Vijjini et al., 2024). Second, current approaches of social intelligence in LLMs often focus on the final outcome and neglect the reasoning process (Hu et al., 2025). While rule-based reinforcement learning with outcome rewards has proven effective in eliciting strong objective reasoning in domains such as mathematics and coding (Guo et al., 2025; Wang et al., 2024), social domain receives little attention. Moreover, existing approaches remain confined to outcome-level supervision, rewarding only the correctness of final answers. This narrow focus incentivises models to adopt superficial heuristics rather than engaging in deeper, contextually grounded social reasoning. In contrast, human social reasoning develops through better processing of social information (Salancik and Pfeffer, 1978) and recursive inference about others' mental states (Frith and Frith, 2005), processes that collectively underpin the emergence of higher levels of social intelligence(Jacobs et al., 2020).

Motivated by the recent success of reinforcement-learning pipelines such as DeepSeek-R1(Guo et al., 2025) and Kimi k1.5(Team et al., 2025) in strengthening general reasoning, we explore whether a similar strategy can unlock the social dimension of cognition. We start with **ToMBench-Hard**, a rigorously vetted Theory-of-Mind benchmark composed of hard cases hand-crafted by human experts to foil shortcut solutions and reveal genuine social reasoning. The benchmark quickly exposes current weaknesses: even state-of-the-art LLMs such as O3 and GPT-5 score below 64%, whereas human annotators exceed 87 %. Fine-tuning models on ToMBench-Hard with a standard outcome-based reward already lifts performance markedly, confirming that reinforcement signals can guide LLMs toward more faithful social inference. Yet a sizable gap to human competence persists, suggesting that answer-level feedback alone is insufficient. To bridge this gap, we collect diverse expert reasoning traces, train a social-thinking reward model that judges each step for accurate perception of social cues and Theory-of-Mind-consistent inference, and blend this trajectory-level feedback with the outcome reward. The resulting framework, Social-R1, sets new records on ToMBench-Hard and transfers robustly to EmoBench, ToMBench, SimpleToM, and MotiveBench; strikingly, an 4 B-parameter Social-R1 model surpasses LLaMA3-70B on every evaluation. These findings highlight that aligning both what a model concludes and how it reasons is critical for cultivating human-like social intelligence in language models.

In summary, our contributions are as follows:

- We propose **ToMBench-Hard**, a 900-sample Theory-of-Mind multiple-choice benchmark that provides a more faithful evaluation of social reasoning abilities with hard samples. Moreover, directly conducting Rule-based GRPO with ToMBench-Hard can already significantly boost LLMs' social intelligence.

- Based on the training set split from ToMBench-Hard, we further construct the SocialReward-3k dataset and train a **social thinking reward model** that evaluates the quality of reasoning trajectories aligned with the social information processing theory.

- We introduce **Social-R1**, a reinforcement learning framework that integrates both outcome-level rewards with trajectory-level thinking rewards to explore how to enhance social reasoning in LLMs in a genuine, robust, and systematic manner.

- We conduct comprehensive experiments with Social-R1, involving both **in-domain** and **out-of-domain** social intelligence-related benchmarks. Results demonstrate that Social-R1 significantly and consistently improves social intelligence across domains.

## 2 RELATED WORK

### 2.1 THEORY-OF-MIND IN LLMS AND LIMITATIONS

A growing body of work evaluates LLMs on ToM tasks, ranging from classical false-belief vignettes to richer, dynamic story settings. Kosinski (2024) shows that recent models such as GPT-4 solve about 75% of false-belief tasks, roughly matching the performance of six-year-old children, whereas earlier models lag far behind. Street et al. (2024) reports that large-scale LLMs like GPT-4 and Flan-PaLM reach (or nearly reach) adult-level performance on a variety of ToM assessments. Strachan et al. (2024) finds that GPT-4 performs at, and sometimes above, human level on false-belief, hinting, irony, and strange-stories tasks, concluding that LLM behavior is often consistent with human mentalistic inference. However, a broader line of literature reveals important weaknesses. Gandhi et al. (2023) notes inconsistent results across studies and therefore proposes a generative evaluation framework based on causal templates to expose systematic failure modes. Chen et al. (2024) offers the most comprehensive ToM audit so far, covering eight tasks and 31 capabilities. Xu et al. (2024) improves test quality by using longer, unstructured narratives, explicit character traits, and action-based evaluation; it shows that LLMs excel at modeling mental states tied to the physical world but fall short in purely psychological contexts, a pattern also observed by Gu et al. (2024); Zhou et al. (2023). Using dynamic scenarios instead of static QA items, Xiao et al. (2025) demonstrates that LLMs lag far behind humans when characters' mental states evolve over time. He et al. (2023) probes higher-order ToM and reveals a sharp performance drop beyond first-order reasoning. Attempts to improve ToM remain limited. Gu et al. (2024) employ system prompts that instruct the model to perform explicit mental reasoning before acting, but this is a task-specific intervention. ToM-RL Lu et al. (2025b) combines reinforcement learning with data generated from Hi-ToM He et al. (2023), achieving encouraging gains; yet without a broader set of challenging training samples, the model may overfit to high-order ToM patterns and fail to generalize to wider facets of social intelligence.

From this literature we distill three limitations that motivate our work: (i) most benchmarks measure only outcome correctness, not the underlying reasoning trajectory; (ii) models often rely on shallow statistical cues instead of genuinely recursive mental-state inference; and (iii) out-of-distribution generalization—both across other ToM benchmarks and to related social-intelligence tasks such as EmoBench Sabour et al. (2024) and MotiveBench (Yong et al., 2025)—remains weak. These limitations point to the need for (a) harder, adversarial ToM evaluations; (b) supervisory signals that target the reasoning process itself (trajectory-level supervision); and (c) training regimes explicitly designed to foster broad, transferable social intelligence.

## 2.2 RL FOR LLM GENERAL REASONING

RL has become a central technique for refining the behavior and reasoning of large language models. RLHF now serves as a standard alignment approach (Ouyang et al., 2022). More recently, large-scale RL with verifiable, outcome-based rewards—drawn directly from task performance—has been shown to markedly strengthen general reasoning abilities. Industry efforts such as OpenAI O1 (Jaech et al., 2024), DeepSeek R1 (Guo et al., 2025), Kimi K1.5 (Team et al., 2025), and Qwen 3 (Yang et al., 2025) confirm that pure RL training can unlock new capabilities, especially on objective tasks requiring structured reasoning and code or math generation. Outcome-based rewards, however, are often sparse and provide poor credit assignments across long reasoning chains. To mitigate this, a complementary line of work explores process supervision, which delivers dense, step-by-step feedback on the reasoning trajectory itself (Lightman et al., 2023; Uesato et al., 2022; Zhang et al., 2025). Process reward models (PRMs), trained on human- or model-generated labels for each intermediate step, can penalize faulty logic even when the final answer is correct, nudging the model toward more reliable reasoning paths.

Although RL methods have significantly improved LLM performance on complex logical tasks such as mathematics and coding, they remain under-explored in the domain of social inference. Social scenarios are typically unstructured and require the model to empathize with others and reason over multiple interacting social factors. A first step in this direction is (Lu et al., 2025b), which applies outcome-based RL using synthetic ToM examples. While this pioneering study shows that RL can, in principle, train social intelligence, it remains in an early stage. These gaps motivate our work: (i) refreshing and expanding training materials to better support RL for social reasoning, and (ii) investigating trajectory-level rewards that guide LLMs toward more human-like social inference.

## 3 METHODOLOGIES OF SOCIAL-R1

In this section, We introduce **Social-R1**, a reinforcement learning-based framework Incentivized over hard ToM samples to enhance social reasoning in LLMs. It mainly contains three components: a challenging social reasoning benchmark ToMBench-Hard (Section 3.1), the social thinking reward design (Section 3.2), and the optimization with RL (Section 3.3).

### 3.1 TOMBENCH-HARD

**Data Construction**   ToMBench-Hard is deliberately curated to address the limitations outlined in Section 2.1, with the goals of ensuring broad coverage across Theory of Mind dimensions, and increasing task difficulty by introducing nuanced distractors and context-dependent social reasoning. Guided by the well-defined psychological framework "Abilities in the Theory-of-Mind Space (ATOMS) " (Osterhaus and Bosacki, 2022), we design multiple-choice questions that probe reasoning across six distinct ability dimensions: *Emotion*, *Desire*, *Intention*, *Knowledge*, *Belief*, and *Non-literal Communication*. Inspired by prior work (Ullman, 2023; Hu et al., 2025), we further increase task difficulty by introducing subtle but ToM-consistent adversarial perturbations—such as manipulations of perceptual access or asymmetric information—that compel models to engage in deeper and more robust reasoning. ToMBench-Hard is manually constructed and annotated by the authors of this paper and other researchers with expertise in natural language processing and cognitive psychology. Each item consists of a social scenario, a question, and multiple candidate options, with only one correct answer. All samples are cross-checked independently by three annotators to ensure quality and consistency. Detailed procedures for data construction and annotation are provided in Appendix A.1.2.

**Benchmarking LLMs with TomBench-hard**   ToMBench-Hard consists of 900 multiple-choice questions spanning six major ToM dimensions. As shown in Table 1, we present both scores for human performance as well as a range of LLMs. Human performance reaches around overall 87% accuracy, whereas both closed- and

open-source LLMs lag considerably behind, scoring below 64% and 51%, respectively. As a comparison, we also perform evaluations with the synthetic dataset from ToM-RL (Lu et al., 2025b), which in fact only covers only the *Belief* dimension. We can observe that O3 and GPT-5 can exceed 87% overall accuracy, and an 8B Qwen3 model can achieve 73% on ToM-RL dataset, compared to less than 48% on our ToMBench-Hard. This performance gap highlights the difficulty posed by ToMBench-Hard and its potentials in deeply incentivizing genuine, human-like social reasoning capabilities rather than capturing shortcuts commonly exist in pretrained datasets. Detailed statistics are provided in Appendix A.1.1.

Table 1: Benchmarking LLMs with **ToMBench-Hard** and **ToM-RL**. ToMBench contains a wide range of hard samples, which is beneficial in inducing robust, intelligent, and human-like social reasoning capabilities.

| | ToMBench-Hard (All) | | | | | | | ToM-RL |
|---|---|---|---|---|---|---|---|---|
| | **Emotion** | **Desire** | **Intention** | **Knowledge** | **Belief** | **Non-literal Communication** | **Overall** | |
| **Human** | 0.8400 | 0.8889 | 0.8128 | 0.9020 | 0.8485 | 0.9385 | 0.8718 | – |
| O3 | **0.7600** | **0.6792** | **0.5047** | 0.5882 | 0.6212 | 0.6792 | **0.6388** | **0.8789** |
| GPT-5 | 0.7200 | 0.5741 | 0.4692 | 0.5098 | **0.6591** | **0.6923** | 0.6041 | 0.8744 |
| GPT-5 + COT | 0.6857 | 0.5741 | 0.4194 | **0.6078** | **0.6591** | **0.6923** | 0.5462 | 0.8322 |
| GPT-4o | 0.6971 | 0.5000 | 0.4171 | 0.3922 | 0.6061 | 0.5846 | 0.5328 | 0.7222 |
| GPT-4o + COT | 0.6971 | 0.4444 | 0.4431 | 0.3922 | 0.6288 | 0.6923 | 0.5497 | 0.6611 |
| Qwen3-4B (Disable thinking) | 0.6057 | 0.3704 | 0.6588 | 0.2941 | 0.5682 | 0.5385 | 0.5059 | 0.5733 |
| Qwen3-4B | 0.5600 | 0.3148 | 0.3981 | 0.1373 | 0.4545 | 0.4615 | 0.3877 | 0.6656 |
| Qwen3-8B (Disable thinking) | 0.6400 | 0.3704 | 0.4834 | 0.3137 | 0.5000 | 0.5538 | 0.4769 | 0.7378 |
| Qwen3-8B | 0.4743 | 0.3148 | 0.3436 | 0.1569 | 0.4318 | 0.3846 | 0.3510 | 0.7333 |
| Qwen3-32B (Disable thinking) | 0.6457 | 0.4630 | 0.4455 | 0.3137 | 0.5303 | 0.7231 | 0.5022 | 0.6900 |
| Qwen3-32B | 0.6514 | 0.3889 | 0.5166 | 0.2745 | 0.5833 | 0.6154 | 0.5050 | 0.7089 |
| LLaMa3.1-70B | 0.5200 | 0.3519 | 0.3270 | 0.3137 | 0.4091 | 0.5385 | 0.4100 | 0.6578 |

## 3.2 SOCIAL THINKING REWARD MODEL

**Thinking Reward Design** One interesting observation from Table 1 is that, conventional Chain-of-Thought (CoT) or advanced thinking processes fail to either consistently nor significantly enhance social reasoning performance. This is probably due to, unlike logical and symbolic reasoning tasks such as math and coding, social tasks require grounding in social information, emphasizing with others, and reasoning embedded in the interaction context. Inspired by the Social Information Processing (SIP) theory (Salancik and Pfeffer, 1978), we design our reward around the stages of the human social reasoning process. According to SIP, an individual in a social situation first perceives and encodes relevant social cues (*perception and encoding stage*). Next, they interpret these cues and develop Theory-of-Mind reasoning toward the social partner (*interpretation stage*), and finally decide which action to take. Based on this framework, we design the reward model along three key dimensions: (1) accurately perceiving and interpreting social cues in scenarios, (2) ensuring ToM reasoning is logically consistent and free of contradictions, and (3) keeping the reasoning concise and without redundancy. Detailed criteria for each dimension are provided in Appendix A.2.

**Thinking Reward Data Collection** To construct high-quality thinking data for reward model training, we first prompt OpenAI o3 to generate raw reasoning trajectories for correct answers and then manually refine

them to ensure accurate recognition of social cues and faithful Theory of Mind reasoning. These refined trajectories serve as the *golden reasoning processes*. We then collect candidate reasoning processes from GPT-4o, Qwen3-8B, Qwen3-32B, and randomly sample an option to force the thinking processes toward the direction to the option. Each candidate reasoning process is scored against the gold reasoning process by GPT-5 on a 0-to-1 scale with increments of 0.1. In total, we obtain 6,300 annotated reasoning trajectories. Next, we construct a set of pairwise data samples for reward model training, basically, each time we draw two samples from the annotated reasoning trajectories, the one with the higher annotated score will serve as the win case the other as the lose case. To ensure balanced quality, we apply rule-based filtering to remove noisy samples and perform uniform sampling across reward intervals, resulting in the **SocialReward-3k** dataset. A social thinking reward model, initialized from Qwen3-4B, is trained on this dataset using supervised fine-tuning. The model is tasked with predicting a scalar reward given a scenario, a question, options, and the corresponding reasoning process. Through this training, the model learns to detect reasoning errors and assign appropriate scores, thereby providing critical feedback on reasoning quality during reinforcement learning.

### 3.3 OPTIMIZATION WITH RL

We adopt the GRPO (Guo et al., 2025) as the optimization method under the RL framework. As illustrated in Figure 2 (a), the process begins with the reasoning model generating a group of different responses $\{o_i\}_{i=1}^{N}$ for each query $q$, where each response includes both reasoning steps and a final answer, enclosed within <think></think> and <Answer></Answer> tags, respectively.

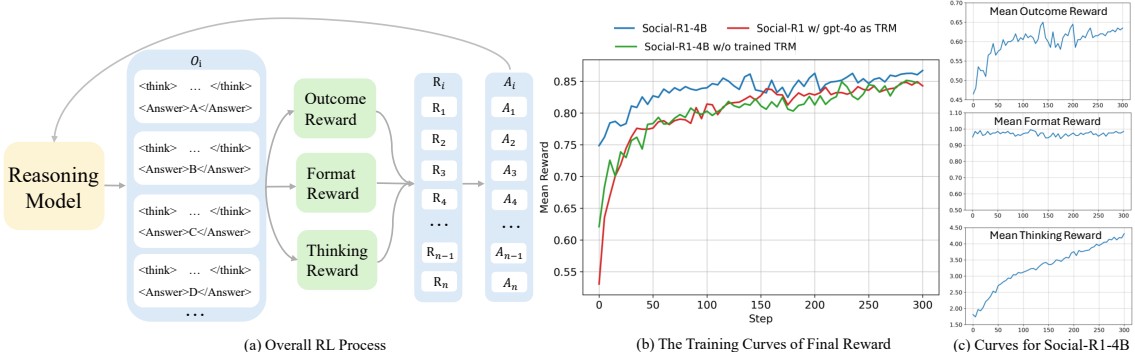

(a) Overall RL Process     (b) The Training Curves of Final Reward     (c) Curves for Social-R1-4B

Figure 2: An illustration of (a) Social-R1 overall framework with GRPO; (b) The training curves of the final merged rewards for three settings: Social-R1 4B, Social-R1 4B w/ gpt-4o as TRM, and Social-R1 4B w/o trained TRM; and (c) the training curves of each individual reward from Social-R1 4B

**Reward** Each response is evaluated from three perspectives: (i) a *format reward*, which ensures the structural correctness of outputs, including both the thinking and answer blocks; (ii) an *outcome reward*, which measures whether the predicted answer matches the ground truth; and (iii) a *thinking reward*, which assesses the quality of intermediate reasoning trajectories based on social cognition principles designed in Section 3.2. The final reward is a linear combination, formulated as:

$$R_i = \frac{\lambda_f R_i^f + \lambda_o R_i^o + \lambda_t \, \sigma(R_i^t)}{\lambda_f + \lambda_o + \lambda_t}, \tag{1}$$

where $R_i^f$, $R_i^o$, and $R_i^t$ denote the format, outcome, and thinking rewards, respectively, $\sigma(\cdot)$ is the sigmoid function, and $\lambda_f$, $\lambda_o$, and $\lambda_t$ are weighting coefficients. Figure 2 (b) and (c) offer the training curves with those rewards.

**Advantage Estimation**   The advantage of each response is normalized within the group to stabilize training, i.e., $A_i = \frac{R_i - \mu_G}{\sigma_G}$, where $\mu_G$ and $\sigma_G$ are the mean and standard deviation of group rewards.

**Policy Optimization**   Following GRPO (Guo et al., 2025), we optimize the policy $\pi_\theta$ by sampling responses from the old policy $\pi_{\text{old}}$ and maximizing the clipped surrogate objective:

$$J_{\text{Social-R1}}(\theta) = \mathbb{E}_{q \sim P(Q), \{o_i\}_{i=1}^N \sim \pi_{\text{old}}} \left[ \frac{1}{N} \sum_{i=1}^{N} \left( \min \left( \frac{\pi_\theta(o_i|q)}{\pi_{\text{old}}(o_i|q)} A_i, \right. \right. \right.$$
$$\left. \left. \left. \text{clip}\left( \frac{\pi_\theta(o_i|q)}{\pi_{\text{old}}(o_i|q)}, 1 - \epsilon, 1 + \epsilon \right) A_i \right) - \beta D_{\text{KL}}[\pi_\theta \parallel \pi_{\text{ref}}] \right) \right],$$

(2)

where $\pi_{\text{ref}}$ is the reference policy and $\beta$ is the KL regularization coefficient.

## 4   EXPERIMENT

### 4.1   EXPERIMENT SETTING

**Benchmarks**   We evaluate our model on six multiple-choice social benchmarks, including two in-domain benchmarks—the public ToMBench (Chen et al., 2024) and our ToMBench-Hard test set — and four out-of-domain benchmarks: SocialIQA (Sap et al., 2019) for social commonsense reasoning, EmoBench (Sabour et al., 2024) for emotion understanding and application evaluation, MotiveBench (Yong et al., 2025) for social motivation reasoning, and SimpleToM (Gu et al., 2024) for examining whether models can consciously infer others' mental states (MS) and proactively applying such reasoning to behavior inference.

**Implementation Details**   The social thinking reward model is trained as a pairwise reward model. We initialize it from Qwen3-4B, use the SocialReward-3k preference dataset containing (chosen, rejected) answer pairs, and finetune with LoRA. The reasoning model is initialized from Qwen3-4B and Qwen3-8B and trained on our ToMBench-Hard dataset, which consists of 700 training samples and 200 test samples. Reinforcement learning is conducted for 300 steps with verl(Sheng et al., 2024) on 16 NVIDIA A100 40GB GPUs. The group size is set to 5, the KL-divergence coefficient to 0.04, and the learning rate to $5 \times 10^{-7}$. More details can be found in Appendix A.3.

### 4.2   MAIN RESULTS

We appy the Social-R1 framework on two open-source models with different sizes: Qwen3-4b and Qwen3-8B. The overall performance is report in Table 2. We can observe that Social-R1 achieves strong and consistent improvements across six social reasoning benchmarks. While closed-source models such as GPT-5 and GPT-4o exhibit strong average performance, Social-R1 shows that reinforcement learning with challenging training data and social thinking reward signals can substantially enhance social reasoning, even when applied to smaller open-source backbones like Qwen3-4B. Notably, Social-R1-4B surpasses LLaMa3-70B on all benchmarks despite the latter having much more parameters; Social-R1-8B surpasses or approaches Qwen-32B model on all the out-of-domain benchmarks.

Another promising observations is, through training, Social-R1 is able to actively leverage social reasoning in practical applications, such as behavioral judgment and emotion understanding. In SimpleToM and EmoBench, Social-R1 models demonstrate substantial improvements in the *behavior & judgment* and *emotion application* dimensions, underscoring their ability to engage in social reasoning and proactively apply it in social applications. Detailed performance of different models across various dimensions in six benchmarks is provided in Appendix A.4.

Table 2: Overall performance of different models across six social reasoning benchmarks. COT indicates Chain-of-Thought. MS indicates Mental State, which is a task-specific prompt-based reminder from (Gu et al., 2024), and it is only applicable for SimpleToM.

| | In-domain | | Out-of-domain | | | |
| --- | --- | --- | --- | --- | --- | --- |
| | ToMBench | ToMBench-Hard (Test set) | SocialIQA | SimpleToM | EmoBench | MotiveBench |
| | | | Closed-sourced LLMs | | | |
| GPT-5 | 0.8168 | 0.5950 | 0.8269 | 0.7355 | 0.8030 | 0.9050 |
| GPT-5+COT | **0.8189** | 0.6000 | 0.8163 | 0.7111 | 0.7882 | 0.9100 |
| GPT-5+MS | – | – | – | **0.9924** | – | – |
| GPT-4o | 0.7769 | 0.5650 | 0.7840 | 0.6661 | 0.5100 | **0.9383** |
| GPT-4o+COT | 0.7899 | 0.5350 | 0.7953 | 0.6126 | 0.7762 | 0.9100 |
| GPT-4o+MS | – | – | – | 0.7358 | – | – |
| | | | Open-sourced LLMs | | | |
| Qwen3-4B (Disable thinking) | 0.6108 | 0.5150 | 0.7313 | 0.5109 | 0.5280 | 0.8550 |
| Qwen3-8B (Disable thinking) | 0.5647 | 0.5000 | 0.7600 | 0.5109 | 0.6484 | 0.8756 |
| Qwen3-32B (Disable thinking) | 0.7318 | 0.5150 | 0.7615 | 0.7239 | 0.6700 | 0.9067 |
| Qwen3-32B | 0.7433 | 0.6000 | 0.7774 | 0.7437 | 0.6356 | 0.9017 |
| LLaMa3.1-70B | 0.6573 | 0.4050 | 0.4621 | 0.7190 | 0.6632 | 0.5633 |
| Qwen3-4B | 0.6402 | 0.3950 | 0.7631 | 0.5411 | 0.5308 | 0.8667 |
| Social-R1-4B | 0.6827 (+6.6%) | 0.6597(+67%) | 0.7736 (+1.4%) | 0.9365 (+73.1%) | 0.6780 (+27.7%) | 0.8709 (+0.5%) |
| Qwen3-8B | 0.6685 | 0.5100 | 0.7871 | 0.6176 | 0.5613 | 0.8822 |
| Social-R1-8B | 0.6875 (+2.8%) | **0.7000** (+37.2%) | 0.7874 (+0.04%) | 0.8963 (+45.1%) | 0.7212 (+28.5%) | 0.8931 (+1.2%) |

## 5 ABLATION STUDY

Next, we conduct ablation studies to quantify the contribution of each component in **Social-R1**, particularly on validating the effectiveness of ToMBench-Hard and the TRM. We evaluate four variants:

- **Social-R1 w/o Hard&TRM**: replace ToMBench-Hard with the training data from ToM-RL (Lu et al., 2025b) and use outcome-only reward, to assess whether our hard training cases are necessary to elicit stronger social reasoning.
- **Social-R1 w/o TRM**: train with outcome-only reward on ToMBench-Hard, to isolate the effect of TRM.
- **Social-R1 w/o trained TRM**: substitute the trained TRM with an untrained Qwen3-4B, to examine the effectiveness of social thinking reward model.
- **Social-R1 w/ gpt-4o as TRM**: replace the TRM with gpt-4o as an generative and general-purpose TRM, to verify whether using a stronger TRM (gpt-4o vs. Qwen3-4B) can make a difference.

As shown in Table 3, Social-R1 overall achieves the best performance across most benchmarks, demonstrating a certain degree of generalized and robust boosted social intelligence. While introducing an untrained reward model still brings overall gains over the baseline, its improvement is not robust across different benchmarks and it falls short of the improvements provided by the trained TRM.

**Effect of Hard Cases**   When ToMBench-Hard is replaced with synthetic ToM-RL data trained under outcome-only rewards (**Social-R1 w/o ToMBench-Hard&TRM**), performance drops substantially across multiple benchmarks. This indicates that the more challenging cases in ToMBench-Hard provide essential supervision signals, guiding the model toward deeper social reasoning. In contrast, synthetic data with relatively easier cases not only fails to elicit comparable reasoning capability but also weaken the model's original social reasoning ability.

**Effect of Thinking Reward Model**   Replacing the trained TRM with an untrained Qwen3-4B (**Social-R1 w/o trained TRM**) weakens performance on all the other benchmarks except SimpleToM, confirming the necessity of training the reward model on social reasoning trajectories (i.e., SocialReward-3k). Using **GPT-4o-as-TRM** leads to unstable or weaker results, especially on SocialIQA, EmoBench and MotiveBench, underscoring that a general-purpose model is less reliable as a reward evaluator than a dedicated social thinking reward model. These findings highlight the importance of tailoring the social thinking reward model to capture social reasoning signals for stable and robust improvements.

Table 3: Ablation results. Results are splitted into two sections by the backbone models, i.e., Qwen3-4B and Qwen3-8B, and the best results in each section are highlighted in green.

| | In-domain | | Out of domain | | | |
|---|---|---|---|---|---|---|
| | **ToMBench** | **ToMBench_Hard** | **SocialIQA** | **SimpleToM** | **EmoBench** | **MotiveBench** |
| Qwen3-4B | 0.6402 | 0.3950 | 0.7631 | 0.5411 | 0.5308 | 0.8667 |
| Social-R1 4B w/o Hard&TRM | 0.6358 | 0.5100 | 0.7699 | 0.5388 | 0.5528 | 0.8583 |
| Social-R1 4B w/o TRM | 0.6455 | 0.6600 | 0.7666 | 0.9718 | 0.5528 | 0.8656 |
| Social-R1 4B w/ gpt-4o as TRM | 0.6309 | 0.6834 | 0.7462 | 0.9718 | 0.6582 | 0.8333 |
| Social-R1 4B w/o trained TRM | 0.6306 | 0.6350 | 0.7533 | 0.9794 | 0.6482 | 0.8644 |
| Social-R1 4B | 0.6658 | 0.6597 | 0.7736 | 0.9365 | 0.6780 | 0.8709 |
| Qwen3-8B | 0.6685 | 0.5100 | 0.7871 | 0.6176 | 0.5920 | 0.8822 |
| Social-R1 8B w/o-TRM | 0.6764 | 0.6850 | 0.7779 | 0.9741 | 0.7205 | 0.8633 |
| Social-R1 8B w/ gpt-4o as TRM | 0.6864 | 0.7052 | 0.7768 | 0.8917 | 0.7036 | 0.8789 |
| Social-R1 8B w/o trained TRM | 0.6863 | 0.6089 | 0.7815 | 0.9756 | 0.5713 | 0.8633 |
| Social-R1 8B | 0.6875 | 0.7000 | 0.7874 | 0.8963 | 0.7212 | 0.8931 |

## 6   CONCLUSION

In this work, we introduce **ToMBench-Hard**, a challenging benchmark that rigorously evaluates the Theory of Mind capabilities in LLMs. Building on this, we propose **Social-R1**, a reinforcement learning framework that integrates both outcome-level and thinking-level rewards to cultivate human-like social intelligence in LLMs. Our results demonstrate that outcome-based reinforcement learning over ToMBench-Hard already enhances social reasoning, while thinking-level supervision yields further improvements. Strikingly, Social-R1-4B surpasses LLaMA3-70B across all evaluated benchmarks, despite the latter having more than an order of magnitude more parameters, underscoring the efficiency of our approach. Together, these findings highlight the importance of supervising not only what a model concludes but also how it reasons, paving the way toward socially intelligent LLMs. Future work may extend this framework to broader domains of social tasks, such as human-AI collaboration, human value alignment and moral reasoning, and LLM-based simulations for social science.

## 7 ETHICS STATEMENT

Our work uses only publicly available benchmark (ToMBench, SocialIQA, EmoBench, MotiveBench, SimpleToM) to evaluation LLMs and newly created data that does not contain personally identifiable information. All annotations were conducted by recruited graduate students with informed consent and fair compensation. This research does not involve human subjects or sensitive personal data. This work complies with the ICLR Ethics Guidelines.

## 8 REPRODUCIBILITY STATEMENT

We provide all necessary resources to ensure the reproducibility of our results. Specifically, (i) detailed experimental settings, including hyperparameters, training configurations, and all prompts used for evaluation, are presented in the Appendix; (ii) an anonymized repository containing the complete codebase for model training and evaluation will be made publicly available, ensuring full transparency[1]; and (iii) a comprehensive README file is included to guide users through environment setup, dataset preparation, and step-by-step reproduction of all reported results. To reduce variance, all results reported in the paper are obtained by averaging over three independent runs.

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

## A   APPENDIX

### A.1   ToMBench_Hard

ToMBench_Hard is deliberately curated to increase task difficulty by introducing nuanced distractors and context-dependent reasoning. Inspired by the Abilities in the Theory-of-Mind Space (ATOMS) framework (Osterhaus and Bosacki, 2022), each question is designed to probe a distinct aspect of ToM reasoning, detailed definition of each subabilities and dimension can be found in (Osterhaus and Bosacki, 2022) . To further increase difficulty (Ullman, 2023; Hu et al., 2025), adversarial variations such as asymmetric access to information, discrepant intentions, and subtle social cues are included.

#### A.1.1   STATISTIC

ToMBench-Hard consists of 900 multiple-choice questions covering six major dimensions of Theory of Mind: Belief, Desire, Intention, Knowledge, Emotion, and Non-literal Communication and 31 ToM subabilities. The detailed distribution is shown in Table 4. The dataset is divided into a training set of 700 samples and a validation set of 200 samples. This split ensures that all six ToM dimensions are represented proportionally across both subsets.

### A.1.2 DATA ANNOTATION DETAILS

ToMBench_Hard is developed jointly by the author and one psychology graduate student, who construct the scenarios, questions, options and answers. Annotation is carried out by five computer science graduate students (after receiving training) and five social psychology graduate students. Each sample is independently answered by two annotators, and disagreements are discussed and resolved through group review and iterative modification. This procedure ensures both linguistic clarity and psychological validity. The annotation process emphasized consistency across dimensions and aimed to capture nuanced aspects of social reasoning.

Table 4: Statistics of ToMBench_Hard across six ToM dimensions and 31 ToM subabilities

| Category | Count |
|---|---|
| Belief | 132 |
|   Beliefs based on perception | 59 |
|   Second-order belief | 21 |
|   Identity false belief | 14 |
|   Location false belief | 18 |
|   Sequence false belief | 9 |
|   Content false belief | 11 |
| Desire | 44 |
|   Desires in conflict | 17 |
|   Desire-action discrepancy | 14 |
|   Multiple desires | 9 |
|   Desire discrepancy | 4 |
| Emotion | 175 |
|   Typical emotion | 85 |
|   Mixed emotion | 24 |
|   Atypical emotion | 23 |
|   Hidden emotion | 16 |
|   Emotion regulation | 14 |
|   Moral emotion | 9 |
|   Emotion discrepancy | 4 |

| Category | Count |
|---|---|
| Intention | 422 |
|   Intention recognition | 215 |
|   Intention prediction | 181 |
|   Complex intention | 14 |
|   Discrepant intention | 12 |
| Knowledge | 62 |
|   Knowledge inference | 15 |
|   Informational knowledge | 10 |
|   Knowledge perception | 13 |
|   Informational knowledge | 4 |
| Non-literal Communication | 65 |
|   Faux pas | 17 |
|   Involuntary lies | 12 |
|   Irony/Sarcasm | 7 |
|   Egocentric lies | 10 |
|   White lies | 5 |
|   Humor | 9 |
| **Total (ToMBench_Hard)** | **900** |

### A.1.3 CASES IN TOMBENCH_HARD

We provide illustrative examples of the six Theory of Mind dimensions covered in ToMBench_Hard (see Figure 3). These cases demonstrate how solving the questions requires models to go beyond surface-level cues and engage in deeper social reasoning.

| | |
|---|---|
| **Ability** | Belief: Location False Beliefs |
| **Story** | Mary and Mike are wandering around the factory. They see a **transparent** box and some cabinets, and they find a hammer on the box. Mike then leaves the factory, and Mary moves the hammer into the box. |
| **Question** | When Mike returns, where does Mary think he believes the hammer is? |
| | A. On the box  B. In the box  C. In the cabinet  D. In the cupboard |
| **Answer** | B. In the box |

| | |
|---|---|
| **Ability** | Desire: Discrepant desires |
| **Story** | James is a driven corporate lawyer who believes that constant career advancement is the true measure of personal achievement. In contrast, Emily is a painter whose days move at an unhurried pace, convinced that life's value lies in expressing creativity and cherishing beauty rather than chasing professional status. This weekend, James plans to invite Emily to an event. |
| **Question** | What type of event does James most likely invite Emily to? |
| | A. Networking dinner with business executives   B. Contemporary art gallery visit |
| | C. Rare book auction    D. Corporate competition judging |
| **Answer** | B. Contemporary art gallery visit |

| | |
|---|---|
| **Ability** | Intention: Intentions explanations |
| **Story** | In the library, Jack is preparing for the exam, and his friend Emily is listening to music. Jack says, The melody of the song is good, and my headphones are also great. Do you want to use them? |
| **Question** | What does Jack really want to say when he says this? |
| | A. He emphasizes that the melody of the song is good.   B. He emphasizes that his headphones are great. |
| | C. He wants Emily to turn down the volume.        D. He wants to join Emily in listening to the song. |
| **Answer** | C. He wants Emily to turn down the volume. |

| | |
|---|---|
| **Ability** | Emotion: Atypical emotional reactions |
| **Story** | Ethan had been feeling unwell and couldn't attend football practice for a week. On the day he returned, he worried that his teammates might have forgotten him. But as soon as he walked onto the field, his friends ran over, cheered, and hugged him tightly. |
| **Question** | What kind of emotion does Subject: Ethan possibly have? |
| | A. Sadness  B. Hopeless  C. Anger   D. Worried  E. Gratitude  F. Embarrassment. |
| **Answer** | E. Gratitude |

| | |
|---|---|
| **Ability** | Knowledge: Knowledge-pretend play links |
| **Story** | In the mysterious underground world of Terra Valley lives a small, mischievous robot named Grimmo. In this realm without sky or celestial bodies, Grimmo has never seen the heavens, nor encountered a human. Yet Terra Valley is a wonder in itself—its walls adorned with bioluminescent fungi and glimmering minerals that sparkle in the dark. One day, Grimmo begins an imitation routine: extending his arms, he slowly spins on his own axis, his movements reminiscent of a spinning top or a graceful dancer twirling in place. |
| **Question** | What is Grimmo possibly imitating? |
| | A. A floating cloud       B. The rotation of a planet |
| | C. A dancer's spiral turn  D. A glowing mushroom spinning in the wind |
| **Answer** | D. A glowing mushroom spinning in the wind |

| | |
|---|---|
| **Ability** | Non-Literal Communication: Humor |
| **Story** | After the sports day, Mia and Zoe see their friend Jack coming off the running track. Jack is so sweaty that his clothes are completely soaked. Mia nudges Zoe and says: "Jack isn't running on the track—he's swimming in the ocean!" |
| **Question** | Why does Mia say this? |
| | A. Mia really misunderstands that Jack is swimming.          B. Mia lies to make Zoe laugh. |
| | C. Mia jokes to exaggerate Jack's sweatiness.       D. Mia jokes to make Zoe laugh. |
| **Answer** | C. Mia jokes to exaggerate Jack's sweatiness。 |

Figure 3: Representative cases from the six ToM dimensions in **ToMBench_Hard**.

## A.2 SOCIAL THINKING REWARD

### A.2.1 SIP-INSPIRED SOCIAL REWARD RUBRIC

Our reward design is inspired by the Social Information Processing (SIP) theory (Salancik and Pfeffer, 1978). According to SIP, social reasoning unfolds in three stages: (i) *perception & encoding* of relevant social cues, (ii) *interpretation* through Theory-of-Mind inference, and (iii) *response selection*. We use the stage into three evaluation dimensions for reasoning quality as shown in Figure 4.

> You are an expert reasoning evaluator. I will give you an input consisting of five elements: [Story], [Question], [Answer], [Gold Reasoning Process], and [Candidate Reasoning Process].
> Your goal is to compare the Candidate Reasoning Process against the Gold Reasoning Process and judge the quality of reasoning. You must ignore the correctness of the final answer and focus only on the reasoning process.
> Evaluation Criteria:
> 1. Correct Reasoning – Does the Candidate Reasoning Process correctly follow the stages of social information processing, consistent with the Gold Reasoning Process?
>     Perception: Are the relevant social cues accurately identified?
>     Interpretation: Are these cues correctly interpreted in relation to the social context?
> 2. Logical Soundness – Is the reasoning logically coherent, internally consistent, and aligned with the Gold Reasoning Process?
> 3. Redundancy – Is the reasoning concise and relevant, avoiding unnecessary repetition or irrelevant details?
> Scoring Rule: Provide a single score from {0, 0.1, 0.2, 0.3, 0.4, 0.5, 0.6, 0.7, 0.8, 0.9, 1.0} based on reasoning quality, where:
> 0.0 → Completely flawed reasoning (no alignment with the Gold Reasoning Process).
> 1.0 → Perfectly sound reasoning (fully aligned and valid).
> Intermediate values → Reflect partial correctness or minor errors (e.g., 0.3 for major flaws, 0.7 for minor issues).
> Be strict: reward strong reasoning and penalize poor reasoning.

Figure 4: Social Reward Rubric

### A.2.2 DATA COLLECTION

To train the social thinking reward model, we curate a dataset named SocialReward-3k. The pipeline is as follows: **(i) Gold trajectories.** We prompt OpenAI o3 with the correct answer to generate initial reasoning processes(see Figure5), then manually refined them to ensure human-like social information processing and faithful ToM reasoning. These serve as "gold" reasoning. **(ii) Candidate trajectories.** We collect reasoning processes from GPT-4o, Qwen3-8B, Qwen3-32B. For each question, models are required to autonomously generate both the reasoning process and the final answer. In addition, to diversify trajectories, we randomly select an alternative option (besides the correct answer) and prompt the model to produce a reasoning path for this option as well. In this way, each instance yields multiple candidate trajectories. **(iii) Scoring.** Each candidate trajectory is evaluated against the gold standard by GPT-5 on a 0–1 scale in increments of 0.1. The judging process follows the predefined rubric (see Figure 4).**(iv) Filtering & balancing.** Noisy or malformed reasoning is removed via rules, and uniform sampling across reward intervals ensures balanced data. This results in **3,000** high-quality trajectories from an initial pool of **6,300**. **(v) Pairwise construction.** For each sample, we generate pairs of trajectories, designating the higher-scored trajectory as the "win" case and the lower-scored trajectory as the "lose" case. We divide the data into four levels according to the GPT-5 scoring scale: scores of 1 and 0.9 correspond to level A, scores of 0.8, 0.7, and 0.6 to level B, scores of 0.5, 0.4, and 0.3 to level C, and scores of 0.2, 0.1, and 0 to level D. Based on these levels, we construct pairwise

comparisons of varying difficulty: *Hard Pairwise* consists of adjacent levels (A–B, B–C, C–D), *Easy Pairwise* is constructed across the most distant levels (A–D), and *Medium Pairwise* is formed by pairs with one-level separation (A–C, B–D).

To get golden reasoning trajectories, we first prompt o3 to generate raw reasoning trajectories for correct answers and then manually refine them to ensure accurate recognition of social cues and faithful Theory of Mind reasoning. To get candidate reasoning trajectories with scores, we first prompt GPT-4o, Qwen3-8B, Qwen3-32B generate raw reasoning trajectories for correct answers and further prompt and randomly sample an option without the correct answer to force the thinking trajectories toward the direction to the option. Each candidate trajectory is scored against the gold reasoning trajectory by GPT-5 on a 0-to-1 scale with increments of 0.1. Across the 700 training samples, we collected a total of 6,300 reasoning trajectories using four model sources: (1) o3 generates reasoning trajectories only for the correct answers. (2) GPT-4o provides both (i) natural reasoning and (ii) incorrect-answer reasoning. (3) Qwen3-8B provides (i) correct-answer reasoning, (ii) natural reasoning, and (iii) incorrect-answer reasoning.(4) Qwen3-32B provides (i) correct-answer reasoning, (ii) natural reasoning, and (iii) incorrect-answer reasoning.

### A.3 EXPERIMENT IMPLEMENT DETAILS

#### A.3.1 TRAINING DETAILS

The social thinking reward model (TRM) is initialised from `Qwen3-4B` and trained using supervised fine-tuning for two epochs on four NVIDIA A100 80GB GPUs. The model is trained with pairwise comparisons and, during inference, predicts a scalar reward conditioned on (*scenario, question, options, reasoning*). We set all three weight coefficients, $\lambda_f$, $\lambda_o$, and $\lambda_t$, to 1.0.

We evaluate multiple LLM families, including Llama-3.1-70B-Instruct(Dubey et al., 2024)[2], Qwen3-4B/8B/32B(Yang et al., 2025)[3], GPT-5-2025-08-07(Ouyang et al., 2022), GPT-4o-2024-08-06 (Achiam et al., 2023), and o3-2025-04-16. We prompt Qwen3-4B/8B/32B in both Thinking and No-Think setting.

---

**Story** :{}
**Question:**{}
**Answer** :{}
For the given question, the correct answer is "B. Contemporary art gallery visit", explain why it answers the question correctly and enclose it in <think></think>.

---

Figure 5: Gold reasoning prompt

## A.4 DETAILS PERFORMANCE ON TOMBENCH, EMOBENCH, MOTIVEBENCH, SIMPLETOM

Table 5: Performance of all models on SimpleToM

| Model | simpletom_behavior | simpletom_judgment | simpletom_mentalstate | Overall |
|---|---|---|---|---|
| GPT-5 | 0.7350 | 0.4908 | 0.9808 | 0.7355 |
| GPT-5_cot | 0.6949 | 0.4551 | 0.9834 | 0.7111 |
| GPT-5_MS | 0.9913 | 0.9887 | 0.9974 | 0.9924 |
| GPT-4o_cot | 0.5684 | 0.3339 | 0.9355 | 0.6126 |
| GPT-4o | 0.5257 | 0.6957 | 0.7768 | 0.6661 |
| GPT-4o_MS | 0.8326 | 0.4010 | 0.9738 | 0.7358 |
| Qwen3-4B(Disable thinking) | 0.4673 | 0.2415 | 0.8239 | 0.5109 |
| Qwen3-4B | 0.4987 | 0.2214 | 0.9032 | 0.5411 |
| Qwen3-8B(Disable thinking) | 0.4673 | 0.2415 | 0.8239 | 0.5109 |
| Qwen3-8B | 0.5711 | 0.3374 | 0.9442 | 0.6176 |
| llama370B | 0.6722 | 0.5196 | 0.9651 | 0.7190 |
| Social-R1-4B w/o TRM | 0.9808 | 0.9965 | 0.9381 | 0.9718 |
| Social-R1-4B w/ gpt-4o as TRM | 0.9381 | 0.9965 | 0.9808 | 0.9718 |
| Social-R1-4B w/o trained-TRM | 0.9608 | 0.9930 | 0.9843 | 0.9794 |
| Social-R1-4B | 0.9886 | 0.9484 | 0.8726 | 0.9365 |
| Social-R1-8B w/o TRM | 0.9320 | 0.9965 | 0.9939 | 0.9741 |
| Social-R1-8B w/ gpt-4o as TRM | 0.7247 | 0.9878 | 0.9625 | 0.8917 |
| Social-R1-8B-w/o-trained-TRM | 0.9442 | 0.9904 | 0.9922 | 0.9756 |
| Social-R1-8B | 0.8004 | 0.9138 | 0.9747 | 0.8963 |

Table 6: Performance across ToM dimensions on ToMBench.

| Model | Belief | Desire | Emotion | Intention | Knowledge | Non-literal communication |
|---|---|---|---|---|---|---|
| **GPT5** | 0.9274 | 0.6611 | 0.7952 | 0.8706 | 0.6332 | 0.8062 |
| **GPT5+cot** | 0.9151 | 0.6833 | 0.7929 | 0.8618 | 0.6401 | 0.8048 |
| **GPT4** | 0.8594 | 0.6278 | 0.7524 | 0.8235 | 0.5779 | 0.7861 |
| **GPT4+cot** | 0.8980 | 0.6278 | 0.7405 | 0.7971 | 0.5675 | 0.8128 |
| **Qwen3-4B(Disable thinking)** | 0.6168 | 0.5444 | 0.6262 | 0.6324 | 0.3322 | 0.7099 |
| **Qwen3-8B(Disable thinking)** | 0.6383 | 0.5000 | 0.6262 | 0.6441 | 0.2595 | 0.5414 |
| **Qwen3-32B (Disable thinking)** | 0.8073 | 0.6167 | 0.7167 | 0.8147 | 0.4394 | 0.7553 |
| **Qwen3-32B** | 0.8560 | 0.6111 | 0.7286 | 0.8324 | 0.4014 | 0.7553 |
| **lama3.1-70B** | 0.7868 | 0.5000 | 0.6286 | 0.6529 | 0.3875 | 0.6658 |
| **Qwen3-4B** | 0.7506 | 0.5722 | 0.6810 | 0.6882 | 0.7273 | 0.6402 |
| **Social-R1-4B w/o TRM** | 0.6939 | 0.5778 | 0.6667 | 0.6912 | 0.5190 | 0.7245 |
| **Social-R1-4B w/o trained-TRM** | 0.7109 | 0.5833 | 0.6643 | 0.6706 | 0.3806 | 0.7737 |
| **Social-R1-4B w/ gpt-4o as TRM** | 0.7177 | 0.5644 | 0.6333 | 0.6735 | 0.3979 | 0.7988 |
| **Social-R1-4B** | 0.7281 | 0.5989 | 0.6866 | 0.7189 | 0.4957 | 0.7666 |
| **Qwen3-8B** | 0.7982 | 0.5889 | 0.6667 | 0.7176 | 0.2664 | 0.6698 |
| **Social-R1-8B w/o TRM** | 0.6976 | 0.5556 | 0.6976 | 0.7912 | 0.5536 | 0.7626 |
| **Social-R1-8B w/o trained-TRM** | 0.7498 | 0.4778 | 0.7071 | 0.7794 | 0.6021 | 0.8020 |
| **Social-R1-8B w/ gpt-4o as TRM** | 0.7611 | 0.5509 | 0.7467 | 0.7467 | 0.5400 | 0.7721 |
| **Social-R1-8B** | 0.7167 | 0.6145 | 0.7167 | 0.7751 | 0.5211 | 0.7809 |

---

[2]https://huggingface.co/collections/meta-llama-3-1
[3]https://huggingface.co/collections/Qwen/qwen3

Table 7: Performance of all models on MotiveBench tasks.

| Model | Amazon | Persona | Blog |
|---|---|---|---|
| **GPT-5** | 0.9533 | 0.9000 | 0.8667 |
| **GPT-5 COT** | 0.9467 | 0.9067 | 0.8800 |
| **GPT-4o** | 0.9667 | 0.9367 | 0.9133 |
| **GPT-4o COT** | 0.9400 | 0.9200 | 0.8600 |
| **Qwen3-4B (Disable thinking)** | 0.8867 | 0.8533 | 0.8267 |
| **Qwen3-4B** | 0.8933 | 0.8633 | 0.8433 |
| **Qwen3-8B (Disable thinking)** | 0.9000 | 0.8733 | 0.8533 |
| **Qwen3-8B** | 0.9200 | 0.8667 | 0.8600 |
| **Qwen3-32B (Disable thinking)** | 0.9467 | 0.9067 | 0.8667 |
| **Qwen3-32B** | 0.9267 | 0.9200 | 0.8400 |
| **LLaMA3-79B** | 0.7333 | 0.5433 | 0.4333 |
| **Social-R1-4B w/o TRM** | 0.8933 | 0.8767 | 0.8267 |
| **Social-R1-4B w/ gpt-4o as TRM** | 0.8667 | 0.8467 | 0.7867 |
| **Social-R1-4B w/o trained TRM** | 0.8933 | 0.8733 | 0.8267 |
| **Social-R1-4B** | 0.9128 | 0.8733 | 0.8267 |
| **Social-R1-8B w/o TRM** | 0.9133 | 0.8767 | 0.8000 |
| **Social-R1-8B w/ gpt-4o as TRM** | 0.9133 | 0.8833 | 0.8400 |
| **Social-R1-8B w/o trained TRM** | 0.8867 | 0.8767 | 0.8267 |
| **Social-R1-8B** | 0.9400 | 0.8792 | 0.8600 |

Table 8: Performance of all models on EmoBench

| Model | Complex Emotions | Emotional Cues | Personal Beliefs and Experiences | Perspective Taking | Interpersonal | Self | Overall |
|---|---|---|---|---|---|---|---|
| GPT5 | 0.8673 | 0.8571 | 0.7946 | 0.7687 | 0.7800 | 0.7500 | 0.8030 |
| GPT5+cot | 0.8571 | 0.7679 | 0.7679 | 0.7761 | 0.7800 | 0.7800 | 0.7882 |
| GPT4 | 0.4286 | 0.4464 | 0.4196 | 0.3955 | 0.7100 | 0.6600 | 0.5100 |
| GPT4+cot | 0.8163 | 0.8214 | 0.7411 | 0.7687 | 0.7800 | 0.7300 | 0.7762 |
| Qwen3-4B(Disable thinking) | 0.5306 | 0.5179 | 0.4643 | 0.4254 | 0.5800 | 0.6500 | 0.5280 |
| Qwen3-8B(Disable thinking) | 0.7041 | 0.7500 | 0.5893 | 0.5373 | 0.5700 | 0.7400 | 0.6484 |
| Qwen3-32B(Disable thinking) | 0.6939 | 0.7321 | 0.5893 | 0.5448 | 0.6800 | 0.7800 | 0.6700 |
| Qwen3-32B | 0.6531 | 0.6429 | 0.5982 | 0.5597 | 0.6600 | 0.7000 | 0.6356 |
| LLama3.1-70B | 0.7143 | 0.8214 | 0.7143 | 0.6194 | 0.5000 | 0.6100 | 0.6632 |
| Qwen3-4B | 0.5612 | 0.5714 | 0.4821 | 0.4403 | 0.5500 | 0.5800 | 0.5308 |
| Social-R1-4B w/o TRM | 0.6531 | 0.6607 | 0.6429 | NaN | 0.6500 | 0.7100 | 0.5528 |
| Social-R1-4B w/o trained TRM | 0.7143 | 0.7143 | 0.6161 | 0.6045 | 0.6200 | 0.6800 | 0.6582 |
| Social-R1-4B w/ gpt-4o as TRM | 0.6633 | 0.6786 | 0.6429 | 0.5448 | 0.6100 | 0.7500 | 0.6482 |
| Social-R1-4B | 0.6771 | 0.6964 | 0.6396 | 0.6260 | 0.7189 | 0.7100 | 0.6780 |
| Qwen3-8B | 0.5714 | 0.6607 | 0.4821 | 0.4179 | 0.7600 | 0.6600 | 0.5920 |
| Social-R1-8B w/o TRM | 0.7551 | 0.7857 | 0.7054 | 0.6567 | 0.6800 | 0.7400 | 0.7205 |
| Social-R1-8B w/o trained TRM | 0.7551 | 0.7321 | 0.7054 | 0.6791 | 0.6200 | 0.7300 | 0.7036 |
| Social-R1-8B w/ gpt-4o as TRM | 0.7879 | 0.7353 | 0.7188 | 0.6509 | 0.5600 | 0.7100 | 0.6938 |
| Social-R1-8B | 0.7917 | 0.7651 | 0.7128 | 0.6615 | 0.6263 | 0.7700 | 0.7212 |

## A.5 DISCLOSURE OF LLM USAGE

We use Large Language Models (LLMs) to assist with writing. Specifically, LLMs is employed to aid in polishing the language and checking grammar and style. All technical content, experimental design, and analysis are conducted by the authors.

## A.6 QUALITATIVE STUDY

Figure 6 gives the qualitative examples showing shortcut suppression and improved social reasoning in Social-R1-4B compared to Qwen3-4B, with GPT-5 still failing under perceptual-access decoupling. Figure 7 provides examples illustrating how shortcut-like behaviours can be reinstated by introducing perceptual-access and asymmetric-information manipulations.

**Story**
Mary and Mike are wandering around the factory. They see a **transparent** box and some **transparent** cabinets, and they find a hammer in the cabinet. Mike then leaves the factory, and Mary moves the hammer into the box.
**Question**
When Mike returns, where does Mary think he believes the hammer is?
A. On the box    B. In the box   C. In the cabinet  D. In the cupboard
**Answer**    B. In the box

**Thinking** Mary and Mike see the hammer in the cabinet. Both Mary and Mike know the hammer is in the cabinet. Mike leaves the factory. Mary then moves the hammer into the transparent box. Mary knows the hammer is now in the box. But Mike did not see this action. Mike's belief: Since he left before the hammer was moved, he still believes the hammer is in the cabinet. Mary's belief about Mike's belief: Mary knows Mike didn't see her move the hammer, so she thinks Mike believes the hammer is still in the cabinet.
<Answer> C. In the cabinet </Answer>

GPT-5+COT

**Thinking** Mike exits the factory before the location of the hammer changes. Mike is unaware of any subsequent changes in the hammer's location. Mary Moves the Hammer: While Mike is away, Mary moves the hammer from the cabinet into the box. Only Mary sees this change. Therefore, Mike remains unaware that the hammer has been moved.
<Answer> C. In the cabinet </Answer>

Qwen3-4B

**Thinking** Mary and Mike are in a factory where both a transparent box and transparent cabinets are present. They find a hammer in the cabinet. Mike leaves, and Mary moves the hammer into the box. The key detail is that both the box and cabinets are transparent, meaning their contents are visible.When Mike returns, he would see the hammer inside the box because the box is transparent. Mike would believe the hammer is in the box. Mary, having moved the hammer, knows this and would assume that Mike, upon returning, would see it in the box.
<Answer> B. In the box</Answer>

Social-R1 4B

Figure 6: Qualitative examples showing shortcut suppression and improved social reasoning in Social-R1-4B compared to Qwen3-4B, with GPT-5 still failing under perceptual-access decoupling. The highlighted colours indicate different stages of social cognition: pink marks perceiving and encoding relevant social cues, green marks interpreting social cues, and blue marks developing social reasoning.

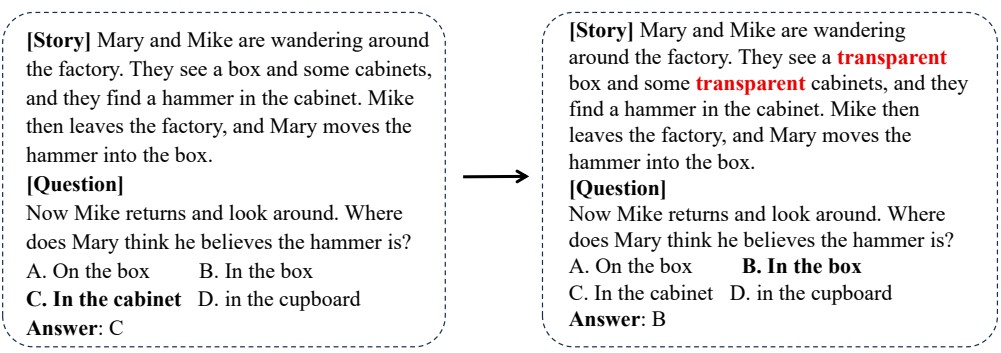

Figure 7: The example show how shortcut-like behaviours can be reinstated by introducing perceptual-access and asymmetric-information manipulations.

