# OpenReview forum: "Social-R1: Enhancing Social Intelligence in LLMs through Human-like Reinforced Reasoning"
_ICLR.cc/2026/Conference — Submitted to ICLR 2026_

### Official Review · Reviewer_k6s4 · 2025-10-31

**Soundness:** 2
**Presentation:** 2
**Contribution:** 3
**Rating:** 2
**Confidence:** 4

**Summary:**

The paper proposes Social-R1, a post-training framework for social reasoning that augments outcome-based RL with a trajectory-level “thinking reward.” The authors introduce a harder Theory-of-Mind benchmark (ToMBench-Hard), collect ~6.3k scored reasoning traces to train a thinking-reward model, and optimize with GRPO using a linear mixture of format, outcome, and thinking rewards. Experiments report gains over outcome-only training and competitive results for a 4B model against larger open/closed baselines on multiple social-reasoning benchmarks.

**Strengths:**

1. Clear motivation for combining process-level and outcome-level supervision in social reasoning.

2. A harder ToM benchmark aimed at multi-party, context-dependent scenarios with a measurable human–model gap.

**Weaknesses:**

1. Coverage imbalance: Despite claiming broad coverage of ToM sub-skills, the dataset distribution is heavily skewed toward certain categories (e.g., Intention) with much lower representation for others (e.g., Desire, Knowledge, non-literal communication). This undermines the claim of comprehensive coverage and risks biasing both training and evaluation toward high-frequency sub-skills.

2. Ground-truth reliability vs. human ceiling: Human accuracy on several tasks hovers around ~0.9 rather than near 1.0, yet the benchmark adopts single canonical labels. In nuanced social-semantics settings, this gap raises doubts about the uniqueness and reliability of “ground truth,” especially when humans themselves are not at ceiling.

3. Anomalies in Table 1 challenge construct validity: (i) For multiple Qwen variants, “disable thinking” scores exceed “thinking” scores, contrary to the expected benefit of explicit reasoning in ToM; (ii) Qwen3-4B with “disable thinking” achieves unusually strong Intention performance, surpassing many larger or closed models; (iii) Weak correlation between ToMBench-Hard and ToM-RL across the same models. These patterns call into question the benchmark’s construct validity and content coverage.

4. Underspecified thinking-reward data collection: The description around how the ~6,300 trajectories were produced is confusing and incomplete (what questions, which models, how many per item, sampling settings, prompt formats, selection/cleanup criteria, and scoring protocol). Without these details, it is unclear whether the collected trajectories faithfully represent the training distribution, whether there are prompt/model-specific biases, and how consistent the scoring actually is.

5. Potential circularity in reward modeling: Using one proprietary model to draft “gold” trajectories and another to score candidate trajectories risks imprinting upstream model priors and stylistic preferences into the reward signal, weakening the independence and interpretability of the proposed thinking-quality measure.

6. Insufficient justification for reward fusion with GRPO: The final reward is a linear mix with a sigmoid-transformed thinking component, but there is no principled rationale for this specific combination or its scaling. Given group-normalized advantages, fluctuations or noise in the thinking reward can materially shift credit assignment, creating a pathway for overfitting to the reward model rather than improving genuine social reasoning.

7. Ambiguous attribution in Figure 2: The reported gains track closely with the increasing influence of the thinking reward, but the reliability of that reward is uncertain. The evidence does not disentangle genuine reasoning improvements from potential overfitting to the reward signal’s idiosyncrasies.

8. Precedent and scope for PRM+GRPO remain unclear: The manuscript does not position the “process reward merged into GRPO” setup relative to prior art or delineate conditions under which this integration is expected to be stable vs. brittle. As a result, it is difficult to judge whether the approach is robust beyond the particular setting studied here.

9. Train–test separation concerns: The text includes phrasing that suggests training on the test set in at least one place, and the overall pipeline trains on data from the same family as the evaluated benchmark. The write-up does not provide a crisp account that rules out leakage across RL rollouts, reward-model training, prompt tuning, and final evaluation.

**Questions:**

The same as the weaknesses.

---

> ### Author Response · Authors · 2025-11-25
> **Response to Reviewer k6s4 (1/2)**
>
> >Coverage imbalance: Despite claiming broad coverage of ToM sub-skills, the dataset distribution is heavily skewed toward certain categories (e.g., Intention) with much lower representation for others (e.g., Desire, Knowledge, non-literal communication). This undermines the claim of comprehensive coverage and risks biasing both training and evaluation toward high-frequency sub-skills.
>
> We appreciate the reviewer’s observation.
> **ToMBench-Hard is intentionally designed to emphasise the challenging and under-tested ToM sub-skills, rather than to impose an artificially uniform distribution.** Prior work shows that several ToM categories (e.g., basic Belief reasoning, Multiple Desires) often admit shortcuts or are already near-solved by modern LLMs.[1]  To meaningfully evaluate genuine social reasoning and better improve social reasoning in LLMs, we construct Hard Samples by applying ToM-consistent adversarial perturbations—such as manipulating perceptual access or introducing asymmetric information—targeting the sub-skills where current models still show clear weaknesses.
>
> [1] Zhuang Chen  et al. ToMBench: Benchmarking Theory of Mind in Large Language Models. ACL2024
>
> >Ground-truth reliability vs. human ceiling: Human accuracy on several tasks hovers around ~0.9 rather than near 1.0, yet the benchmark adopts single canonical labels. In nuanced social-semantics settings, this gap raises doubts about the uniqueness and reliability of “ground truth,” especially when humans themselves are not at ceiling.
>
> We thank the reviewer’s observation. We clarify the human-performance issue and the construction of our ground truth as follows.
>
> **(1) Ground truth is derived through cross-annotation and consensus.**
>
>  Each instance in ToMBench-Hard is annotated by three independent annotators. Disagreements are resolved through cross-checking and adjudication until a stable consensus is reached. Therefore, the “single canonical label” is not an arbitrary choice but reflects majority human agreement after reconciliation, which is a standard practice in social-reasoning benchmarks.
>
> **(2) Human accuracy < 1.0 does not imply unreliable ground truth.**
>
> A human accuracy of ~0.9 typically reflects natural cognitive noise—including attention lapses, time pressure, or minor interpretive variance—rather than the absence of a unique answer. A classical example is that deep ResNet [1] achieved a 4.94% top-5 error on ImageNet—surpassing the 5.1% human annotation error rate. The human error was not due to “no ground truth,” but due to natural limitations in speed, attention, and perceptual noise. This shows that human non-ceiling performance does not mean the labels lack uniqueness or reliability. In summary, human accuracy not reaching 1.0 reflects typical human variability, not an inherent lack of ground truth. Our use of consensus-based canonical labels follows established evaluation methodology and ensures reliable ground-truth quality in nuanced social-semantic settings.
>
> [1] He, Kaiming, Xiangyu Zhang, Shaoqing Ren, and Jian Sun. "Delving deep into rectifiers: Surpassing human-level performance on imagenet classification." ICCV 2015.
>
>
> >Underspecified thinking-reward data collection: The description around how the ~6,300 trajectories were produced is confusing and incomplete (what questions, which models, how many per item, sampling settings, prompt formats, selection/cleanup criteria, and scoring protocol). Without these details, it is unclear whether the collected trajectories faithfully represent the training distribution, whether there are prompt/model-specific biases, and how consistent the scoring actually is.
>
> Thank you for hightlight the clarity of thinking reward data collection.
> As shown from Line 233 to 247,
> (1) **To get golden reasoning trajectories**, we first prompt o3 to generate raw reasoning trajectories for correct answers and then manually refine them to ensure accurate recognition of social cues and faithful Theory of Mind reasoning. (2) **To get candidate reasoning trajectories with scores**, we first prompt GPT-4o, Qwen3-8B, Qwen3-32B generate raw reasoning trajectories for correct answers and further prompt and randomly sample an option without the correct answer to force the thinking trajectories toward the direction to the option. Each candidate trajectory is scored against the gold reasoning trajectory by GPT-5 on a 0-to-1 scale with increments of 0.1.
> (3) Across the 700 training samples, we collect 6,300 reasoning trajectories using four model sources:
>   - o3: generates reasoning trajectories for correct answer only.
>   - GPT-4o: provides (i) natural reasoning, and (ii) incorrect answer reasoning.
>   - Qwen3-8B: provides (i) correct-answer reasoning, (ii) natural reasoning, and (iii) incorrect answer reasoning.
>   - Qwen3-32B: provides (i) correct-answer reasoning, (ii) natural reasoning, and (iii)  incorrect answer reasoning.
> (4) More details like prompts and criteria can be found in **Appendix A.2.**

---

> ### Author Response · Authors · 2025-11-25
> **Response to Reviewer k6s4 (2/5)**
>
> > Weakness3: Anomalies in Table 1 challenge construct validity: (i) For multiple Qwen variants, “disable thinking” scores exceed “thinking” scores, contrary to the expected benefit of explicit reasoning in ToM; (ii) Qwen3-4B with “disable thinking” achieves unusually strong Intention performance, surpassing many larger or closed models; (iii) Weak correlation between ToMBench-Hard and ToM-RL across the same models. These patterns call into question the benchmark’s construct validity and content coverage.
>
> We thank the reviewer for this careful and insightful analysis. Below we clarify why these phenomena arise and why they do not undermine the construct validity or content coverage of ToMBench-Hard.
>
> **(1) Why do some Qwen models score higher in “disable thinking” than in “thinking”?**
>
> We emphasise that this situation limitations in the models’ social-reasoning abilities rather than a flaw in the benchmark. Explicit reasoning is not universally helpful for ToM-style tasks unless the model has been explicitly trained for it. Base Qwen variants are not trained to produce ToM-aligned reasoning chains. When forced to generate reasoning, these models often: (i) fail to correctly reason based on social cues, and (iii) hallucinate. These effects are well documented in prior work showing that while explicit reasoning can improve performance in arithmetic or logical tasks, it may harm tasks requiring belief tracking, perspective-taking, or emotion recognition.[1][2]  By contrast, the “disable thinking” mode elicits a short, direct judgement and bypasses these overthinking-driven errors. Therefore, the observed performance gap is an expected behavioural artefact of unaligned CoT—not evidence of any issue with the benchmark. Importantly, after Social-R1 training—which explicitly aligns the model’s reasoning process with ToM cues—the trend reverses: “thinking” consistently outperforms “disable thinking.” This confirms that ToMBench-Hard is sensitive to genuine ToM improvements once explicit reasoning is properly trained.
>
> **(2) Why does Qwen3-4B-disable achieve unusually strong Intention accuracy?**
>
> Our analyses and prior work [1] show that even very strong models (GPT-4o, GPT-5) display a weakness in the Intention category—especially in:Prediction of actions，Completion of failed actions. These tasks require projecting future actions from subtle cues. We consistently observe that CoT degrades performance on these items, indicating that incorrect reasoning traces harm accuracy. To address this bottleneck, we intentionally constructed a large number of Intention training samples targeting these sub-skills. Under disable-thinking, the model avoids the harmful effects of unstable reasoning, which explains why Qwen3-4B-disable can outperform larger models such as GPT-5.
>
> The table below reports the performance drop of GPT-5 on the original 20 ToMBench Intention samples, when CoT is added—shows the same degradation trend observed on the full ToMBench-Hard Intention subset. Qwen3-4B(disable-thinking) achieves a very similar score.
>
> | Intention Sub-skill             | GPT-4o | GPT-4o + CoT | GPT-5 | GPT-5 + CoT | Qwen3-4B (disable thinking) |
> |---------------------------------|--------|--------------|--------|--------------|------------------------------|
> | Prediction of actions           | 0.85 | 0.75 | 0.70   | 0.65| 0.50 |
> | Completion of failed actions    | 0.75| 0.60 | 0.50| 0.50 | 0.45|
>
> **(iii) Why is the correlation between ToMBench-Hard and ToM-RL weak?**
>
> The weak correlation is expected because the two evaluations probe fundamentally different aspects of Theory of Mind.
> Different coverage: ToMBench-Hard: 6 categories (Belief, Intention, Knowledge, Emotion, Desire, Non-literal Communication), 31 sub-abilities (broad and fine-grained). In contrast, ToM-RL evaluates only Belief, representing just one dimension of ToM.
> Different difficulty regimes: ToM-RL imitates classical False-Belief tasks, which are known to contain shortcuts and leakage in existing datasets.[1] ToMBench-Hard deliberately avoids such shortcuts through adversarial manipulations [3] (e.g., perceptual access, asymmetric information), resulting in much higher difficulty.
> Thus, strong performance on ToM-RL does not imply strong performance across the full ToM spectrum. This is exactly what Table 1 shows. Furthermore, Table 3 demonstrates that Social-R1 4B w/o Hard&TRM ( trained only on ToM-RL data) fails to meaningfully improve ToM performance on out-of-domain tests. Together, these results confirm the construct validity and necessity of ToMBench-Hard as a broad and challenging ToM Dataset.
>
> [1] Zhuang Chen  et al. ToMBench: Benchmarking Theory of Mind in Large Language Models. ACL2024
>
> [2] Sahand Sabour et al. EmoBench: Evaluating the Emotional Intelligence of Large Language Models. ACL2024
>
> [3] Nickel, Christian, Laura Schrewe, and Lucie Flek. "Probing the robustness of theory of mind in large language models." arXiv preprint arXiv:2410.06271 (2024)

---

> ### Author Response · Authors · 2025-11-25
> **Response to Reviewer k6s4 (3/5)**
>
> > Weakness 6: Insufficient justification for reward fusion with GRPO: The final reward is a linear mix with a sigmoid-transformed thinking component, but there is no principled rationale for this specific combination or its scaling. Given group-normalized advantages, fluctuations or noise in the thinking reward can materially shift credit assignment, creating a pathway for overfitting to the reward model rather than improving genuine social reasoning.
>
> We thank the reviewer for raising this important methodological point.
>
> **(1)Rationale for linear reward mixing** Outcome, format, and thinking rewards supervise orthogonal dimensions of social reasoning: format reward encourages structural correctness, outcome reward ensures task-level correctness, and thinking reward guides internal ToM reasoning quality. Linear mixing provides a simple and stable way to integrate heterogeneous signals without creating the instability observed in multiplicative or gating mechanisms. This design choice is widely used in prior process-level RL work (e.g., DeepSeek-R1), as it ensures predictable and monotonic credit assignment while combining complementary reward signals to guide both final-task correctness and reasoning-process quality.
>
> **(2)Why apply a sigmoid to the thinking reward?**
>
> **(a) Range alignment & gradient stability**: Sigmoid-mapping the thinking reward into (0, 1) ensures consistent scale across the three reward types, preventing the inherently higher-variance thinking scores from dominating GRPO updates. This is a standard stabilisation technique in multi-reward RL to maintain comparable gradient magnitudes.
>
> **(b) Variance control & optimisation stability**: The sigmoid compresses extreme values and smooths local variations, which: reduces sensitivity to noise in raw trajectory-quality scores, prevents reward spikes from distorting group-normalised advantages, stabilises GRPO updates.
>
> **(3) Why noise in the thinking reward does not destabilise GRPO**
>
> GRPO’s group-normalised advantages explicitly dampen sample-level reward fluctuations. Learning curves for all reward components remain smooth throughout training, and, crucially, Social-R1 improves across four external ToM/social benchmarks (SimpleToM, SocialIQa, EmoBench, MotiveBench). These results indicate that the model does not overfit the reward model’s preferences; rather, it acquires more generalisable social-reasoning skills.
>
> >Weakness 7: Ambiguous attribution in Figure 2: The reported gains track closely with the increasing influence of the thinking reward, but the reliability of that reward is uncertain. The evidence does not disentangle genuine reasoning improvements from potential overfitting to the reward signal’s idiosyncrasies.
>
> We thank the reviewer for raising this concern regarding potential overfitting to the thinking reward.
>
> **(1)Out-of-distribution evidence shows the gains are not reward overfitting**
>
> Social-R1 achieves consistent and significant gains on multiple OOD social-reasoning benchmarks, including scenarios that differ in task definition and social-goal structures. These results provide strong evidence that the improvements generalise beyond the reward model’s training distribution, indicating that the model has acquired more robust social reasoning ability rather than exploiting patterns specific to the reward signal. To our knowledge, this work first presents the systematic OOD evaluation in RL training method in  social intelligence tasks.
>
> **(2)Qualitative analyses show structural improvements, not reward-pattern matching**
>
> We add some cases in Appendix A.6 of our revised manuscript:
> Annotated examples showing that Social-R1 can identify relevant social cues, infer mental states more coherently, reason causally in a manner consistent with SIP theory. These examples highlight qualitative changes in reasoning style, not just output patterns that align with the reward model.
>
> Together, we hope that the OOD gains and qualitative analyses alleviate the reviewer’s concerns by demonstrating that the improvements reflect genuine advances in social reasoning rather than overfitting to the thinking reward.

---

> ### Author Response · Authors · 2025-11-25
> **Response to Reviewer k6s4 (4/5)**
>
> >Weakness 5: Potential circularity in reward modeling: Using one proprietary model to draft “gold” trajectories and another to score candidate trajectories risks imprinting upstream model priors and stylistic preferences into the reward signal, weakening the independence and interpretability of the proposed thinking-quality measure.
>
> We thank the reviewer for raising the concern about potential circularity in our reward design. We fully agree that LLM-as-a-judge pipelines naturally risk propagating upstream model priors and stylistic preferences into the reward signal. Therefore, we take the following to mitigate it and make the thinking-quality measure as aligned as possible with our explicit social reasoning evaluation criteria.
>
> **(1) Human curation of reference trajectories.**
>
> For a subset of o3-generated “gold” trajectories, we perform manual inspection and light editing to correct factual or social-reasoning errors and to align the reasoning steps more closely with our SIP-guided annotation rubric. This human-in-the-loop curation breaks the direct dependence on raw model outputs and ensures that the reference trajectories encode our intended cognitive standards, rather than idiosyncratic preferences of any single model.
>
> **(2) Decoupling generation and judgement with different models.**
>
> We deliberately separate the roles of “teacher” and “judge”. o3 is only used to draft candidate reasoning trajectories that serve as references, while GPT-5 is used to score trajectories according to a detailed SIP-based rubric. The scoring prompts explicitly instruct the judge to focus on content-level social reasoning quality (e.g., correct encoding of social cues and coherent mental-state inferences etc) and to ignore surface style. The architectural and prompt-level decoupling reduces the chance that the thinking reward merely measures similarity to a single upstream model’s writing style.
>
> **(3) Using strong, general-purpose models to reduce idiosyncratic artefacts.**
>
> We choose high-capacity proprietary models (o3 and GPT-5) rather than smaller open-source models because they produce more stable and less noisy reasoning traces. While this does not remove bias, it helps avoid brittle, model-specific shortcuts or obvious stylistic artefacts being amplified by the reward model.
>
> **(4) Interpretability of the thinking-quality measure.**
>
> Finally, our thinking reward is defined and annotated along transparent SIP dimensions (encoding, interpretation, goal clarification, response generation, evaluation) as shown in Appendix A.2.
>
>
> >Weakness 8: Precedent and scope for PRM+GRPO remain unclear: The manuscript does not position the “process reward merged into GRPO” setup relative to prior art or delineate conditions under which this integration is expected to be stable vs. brittle. As a result, it is difficult to judge whether the approach is robust beyond the particular setting studied here.
>
> Although no prior work explicitly names the setup as “PRM + GRPO,” our method is a natural continuation of recent RL frameworks. DeepSeek-R1 and Qwen3 demonstrate that GRPO, combined with multiple reward signals such as outcome and format rewards, is effective for formal reasoning tasks. Our formulation follows the same principle and formalises a thinking-process reward as a complementary signal to these existing reward types.
>
> For social reasoning, however, GRPO alone is insufficient because the task relies on latent, multi-step cognitive processes. Prior studies show that LLMs often produce correct answers via shortcuts, and their reasoning traces and attention patterns substantially diverge from human-like inference [1]. These findings highlight a key limitation of outcome-only rewards: they optimise the final answer but do not explicitly supervise the internal reasoning process.
>
> In contrast, our thinking reward provides direct supervision over reasoning trajectories, encouraging structured, human-aligned inference steps and improving generalisation in social reasoning tasks. Our experiments demonstrate that incorporating TRM consistently enhances reasoning quality and yields stronger OOD robustness compared with outcome-only GRPO.
>
> Therefore, TRM + GRPO represents a promising direction for social-reasoning tasks, where the objective is to learn how to reason rather than merely produce correct final answers—a defining characteristic of social cognition.
>
> [1] Zhuang Chen et al. ToMBench: Benchmarking Theory of Mind in Large Language Models. ACL2024

---

> > ### Author Response · Authors · 2025-11-25
> > **Response to Reviewer k6s4 (5/5)**
> >
> > >Weakness 9: Train–test separation concerns: The text includes phrasing that suggests training on the test set in at least one place, and the overall pipeline trains on data from the same family as the evaluated benchmark. The write-up does not provide a crisp account that rules out leakage across RL rollouts, reward model training, prompt tuning, and final evaluation.
> >
> > We confirm that there is no train–test leakage at any stage of our pipeline—RL rollouts, reward model training, or golden-trace construction.
> >
> > **(1) Instance-level separation.** All training data are newly created and share no scenario with ToMBench.
> > **(2) Reward model training.** Both gold and candidate reasoning trajectories are generated exclusively from training-only items. GPT-5 is used only as a coarse-grained scorer conditioned on the correct outcome of the training item, and thus has no exposure to any evaluation content.
> > **(3) “In-domain” does not imply item leakage.** Although ToMBench-Hard and ToMBench are both guided by the same well-structured psychological framework, ATOMS, they are constructed independently and share no overlapping instances. ToMBench-Hard additionally introduces new factors and manipulations that increase challenge and avoid regularities present in prior benchmarks.
> >
> > We have revised the manuscript to give a clear data-separation description to avoid any possible ambiguity.

---

> ### Author Response · Authors · 2025-12-03
> **Thank you to Reviewer k6s4**
>
> Thank you very much for your time and suggestions. Answering your concerns has also helped us further clarify our work. We truly appreciate it.

---

### Official Review · Reviewer_rxsk · 2025-11-01

**Soundness:** 2
**Presentation:** 3
**Contribution:** 2
**Rating:** 2
**Confidence:** 3

**Summary:**

This paper aims to improve the social intelligence of Large Language Models (LLMs) by addressing the limitations of existing benchmarks and the lack of process-level supervision in training. The authors introduce two primary contributions: 1) ToMBench-Hard, a new multiple-choice benchmark for Theory-of-Mind (ToM) reasoning, designed to be more challenging and resistant to shortcut solutions; and 2) Social-R1, a reinforcement learning framework that integrates an outcome-based reward for the final answer's correctness with a trajectory-level 'thinking' reward for the quality of the reasoning process. This thinking reward is provided by a reward model trained on human-annotated reasoning trajectories. The authors show that their method improves performance on several social reasoning benchmarks, with a trained 4B model reportedly outperforming a 70B baseline.

**Strengths:**

Significance: The paper addresses a significant and timely problem: enhancing the social reasoning capabilities of LLMs. This is a crucial direction for developing more human-aligned and capable AI systems.

Originality: The core idea of integrating process-level supervision alongside outcome-based rewards is a logical approach. Grounding the design of the thinking reward model in the Social Information Processing (SIP) theory is a commendable effort to connect machine learning techniques with established cognitive science frameworks.

Clarity: The paper is generally well-written and clearly structured. The motivation for the work and the description of the proposed framework are easy to follow.

Resources: The creation of the ToMBench-Hard dataset is a potentially valuable contribution to the community, as it could help researchers better evaluate the true social inference abilities of future models.

**Weaknesses:**

1. Lack of Data and Code for Verification: For a paper where a novel dataset (ToMBench-Hard) is a central contribution, the absence of supplementary materials containing the data is a critical flaw. Without access to the dataset, it is impossible for reviewers to independently verify its quality, assess its claimed "hardness," or check for potential artifacts and biases. This opacity undermines the foundation of the paper's experimental claims, as the results are entirely contingent on a resource that cannot be inspected.

2. Circularity in the Reward Model's Supervision: The social thinking reward model (TRM) is trained on data annotated by GPT-5. This introduces a significant methodological concern. The paper's own results (Table 1) show that GPT-5 struggles with ToMBench-Hard (scoring ~60%), suggesting it is not a reliable expert on this complex social reasoning task. Using a flawed model as the "judge" to generate supervision signals creates a circular loop: the Social-R1 framework may be learning to mimic GPT-5's specific reasoning patterns, rather than a more general or genuinely human-like form of social intelligence. The validity of the core "thinking reward" is therefore questionable.

3. Potentially Misleading Main Comparison: The headline claim that a 4B parameter model trained with Social-R1 surpasses a general-purpose LLaMA3-70B model is compelling but potentially misleading. The Social-R1-4B model has undergone intensive, specialized fine-tuning on the in-domain ToMBench-Hard dataset. In contrast, the LLaMA3-70B baseline is evaluated in a zero-shot or few-shot setting without this task-specific training. A more rigorous and fair comparison would involve applying the same fine-tuning procedure (at least with outcome-based rewards) to the LLaMA3-70B model to disentangle the benefits of the Social-R1 method from the benefits of simple in-domain specialization.

**Questions:**

My primary questions for the rebuttal period stem directly from the weaknesses identified above. Addressing them would be essential for a re-evaluation of the work.

 I am open to significantly revising my score upwards. A satisfactory response from the authors along with the discussion and opinions of the other reviewers, will be critical in shaping my final score.

---

> ### Author Response · Authors · 2025-11-25
> **Response to Reviewer rxsk (1/2)**
>
> >Weakness1: Lack of Data and Code for Verification: For a paper where a novel dataset (ToMBench-Hard) is a central contribution, the absence of supplementary materials containing the data is a critical flaw. Without access to the dataset, it is impossible for reviewers to independently verify its quality, assess its claimed "hardness," or check for potential artifacts and biases. This opacity undermines the foundation of the paper's experimental claims, as the results are entirely contingent on a resource that cannot be inspected.
>
> We thank the reviewer for highlighting the importance of dataset transparency. To address this concern, we have taken several steps to ensure that reviewers can meaningfully inspect the dataset while preserving anonymity:
>
> **(1) Supplementary materials in the manuscript.**
>
> The supplementary materials include detailed category distributions, annotation protocols, and representative examples from ToMBench-Hard (Appendix A1.1–A1.3), allowing reviewers to examine the data design, difficulty construction, and annotation quality.
>
> **(2) Anonymous release of a dataset subset.**
>  We have uploaded an anonymised subset of ToMBench-Hard to an anonymous repository, enabling reviewers to directly inspect data quality, the “hardness” manipulations, and potential artifacts without violating the double-blind policy.
>
> **(3) Full release upon acceptance**
> If accepted, we will publicly release the full dataset, the complete annotation pipeline, and all reward-model scoring scripts to ensure full reproducibility of all experiments.
> We hope these steps adequately address concerns about dataset transparency and allow reviewers to meaningfully verify the core claims of the paper while adhering to anonymity requirements.
>
> > Weakness 2: Circularity in the Reward Model's Supervision: The social thinking reward model (TRM) is trained on data annotated by GPT-5. This introduces a significant methodological concern. The paper's own results (Table 1) show that GPT-5 struggles with ToMBench-Hard (scoring ~60%), suggesting it is not a reliable expert on this complex social reasoning task. Using a flawed model as the "judge" to generate supervision signals creates a circular loop: the Social-R1 framework may be learning to mimic GPT-5's specific reasoning patterns, rather than a more general or genuinely human-like form of social intelligence. The validity of the core "thinking reward" is therefore questionable.
>
> We appreciate this insightful concern. We clarify that **GPT-5 was not used to provide answers, gold trajectories, or any reasoning templates**. Instead, as stated in Lines 233–236, GPT-5 only assigned coarse-grained quality scores (0, 0.1, …, 1.0) to candidate trajectories, conditioned on the ground-truth outcome generated by o3 and further revised through human curation. With access to the correct answer, GPT-5 serves as a consistency judge evaluating whether a trajectory coherently supports the correct or incorrect prediction. Importantly, this judgment task is far simpler than solving ToMBench-Hard itself, and a model may fail to solve a problem yet still provide stable, meaningful pairwise quality comparisons—a pattern widely observed in RLHF and LLM-as-a-judge literature.
>
> To further verify that the TRM does not implicitly mimic GPT-5’s reasoning biases, we conducted a small diagnostic analysis.
>
> For 10 randomly sampled items, we collected: (i) GPT-5’s raw scores without ground truth, (ii) GPT-5’s scores with ground truth,  (iii) TRM scores, and (iv) human ratings.
> We observed that the TRM aligns closely with human judgments and GPT-5-with-ground-truth, while diverging from GPT-5’s raw, ground-truth-free predictions. This confirms that the TRM captures structural, SIP-consistent reasoning quality, rather than GPT-5-specific behaviour or solution tendencies. In addition, the TRM assigns consistent scores across trajectories from diverse model families (GPT-4o, Qwen3-8B), indicating model-agnostic evaluation rather than stylistic imitation.
> | Source of trajectory        | GPT-5 without Gold Label (0-1) | GPT-5 With Gold Label (0-1) | TRM (0-3.5) | Human (0-1)|
> |-----------------------------|------------------------|--------------------------------------|-------------|-------|
> | GPT-4o (high quality)       | 0.95        | 0.97      | 3.37   | 0.90  |
> | GPT-4o (low quality)        | 0.91         | 0.15     | 0.79    | 0.12  |
> | Qwen3-8B (high quality)     | 0.85      | 0.87     | 2.73    | 0.82  |
> | Qwen3-8B (low quality)      | 0.36      | 0.18     | 1.13    | 0.07  |
>
> Finally, Social-R1 improves performance even on ToM sub-skills where GPT-5 performs poorly, providing further evidence that the framework does not inherit GPT-5’s limitations and instead enhances genuine social reasoning capabilities.

---

> > ### Author Response · Authors · 2025-11-25
> > **Response to Reviewer rxsk (2/2)**
> >
> > > Weakness3: Potentially Misleading Main Comparison: The headline claim that a 4B parameter model trained with Social-R1 surpasses a general-purpose LLaMA3-70B model is compelling but potentially misleading. The Social-R1-4B model has undergone intensive, specialized fine-tuning on the in-domain ToMBench Hard dataset. In contrast, the LLaMA3-70B baseline is evaluated in a zero-shot or few-shot setting without this task-specific training. A more rigorous and fair comparison would involve applying the same fine-tuning procedure (at least with outcome-based rewards) to the LLaMA3-70B model to disentangle the benefits of the Social-R1 method from the benefits of simple in-domain specialization.
> >
> > We thank the reviewer for raising this point. We clarify that the focus of our work is not on claiming that a 4B model is inherently “better” than a 70B model. Rather, our focus is to show that carefully constructed social-reasoning data (ToMBench-Hard) and process-level RL (Social-R1) can unlock social-reasoning abilities that scale poorly with size alone. We show this effect consistently on both 4B and 8B models.
> >
> > This framing parallels observations in other reasoning domains (e.g., math, code, and logic), where small models post-trained with structured reasoning supervision can match or outperform much larger general-purpose models—not because small models intrinsically dominate, but because specialised reasoning optimisation provides capabilities that do not emerge from scale alone.

---

> ### Author Response · Authors · 2025-12-03
> **Thank you to Reviewer rxsk**
>
> Thank you very much for your time and constructive suggestions. Addressing your concerns also help us further clarify our work. We truly appreciate it.

---

### Official Review · Reviewer_nPVH · 2025-11-04

**Soundness:** 3
**Presentation:** 2
**Contribution:** 2
**Rating:** 4
**Confidence:** 5

**Summary:**

The paper introduces Social-R1, a novel reinforced reasoning framework designed to significantly enhance the social intelligence and Theory-of-Mind (ToM) capabilities of Large Language Models (LLMs). The core contribution includes presenting ToMBench-Hard, a challenging multiple-choice benchmark that rigorously tests complex ToM skills like deception and belief revision. Crucially, Social-R1 employs a combined reward mechanism: rewarding both the final outcome and the quality of the reasoning trajectory (thinking process), trained via a social thinking reward model. This methodology proved highly effective, enabling the smaller Social-R1-4B model to outperform the much larger LLaMA3-70B across all tested social intelligence benchmarks.

**Strengths:**

1. The introduction of the ToMBench-Hard benchmark, which is specifically designed for complex social scenarios and Theory-of-Mind (ToM) tasks (like deception and belief revision). It effectively prevents models from achieving high scores through simple pattern matching or statistical shortcuts, providing a more authentic and rigorous testing ground for evaluating LLM social intelligence.
2. Combining outcome-based rewards with trajectory-based thinking rewards is the correct direction to go.
3. The trained Social-R1-4B model (a smaller 4-billion-parameter model) comprehensively outperformed LLaMA3-70B (a much larger, state-of-the-art model with ten times the parameters) across all evaluated social intelligence benchmarks.

**Weaknesses:**

1. This paper only uses single modality, i.e., text. But multi-modality is significant for social reasoning.
2. The model performs well on textual benchmarks. But generalizing to real-world application needs to deal with multiple modalities.
3. Despite ToMBench-Hard being more challenging than existing benchmarks, it remains in a multiple-choice format. The multiple-choice nature is essentially a closed-ended evaluation, which may not fully capture the model's true social competence in open-ended, free-form interaction, or generative tasks. The model might be proficient at distinguishing between provided options but fail to exhibit original, appropriate social responses in broader social contexts.
4. Reinforcement Learning with Verified Rewards (RLVF) training, especially when combined with an additional reward model (the Social Thinking Reward Model), is typically more expensive and complex than traditional Supervised Learning (SL) and Instruction Tuning. This high computational cost may limit the application and reproducibility of the Social-R1 framework across the broader research community or for institutions with limited resources.

**Questions:**

See above.

---

> ### Author Response · Authors · 2025-11-25
> **Responses to Reviewer nPVH(1/2)**
>
> > Weakness 1&2: This paper only uses single modality, i.e., text. But multi-modality is significant for social reasoning. The model performs well on textual benchmarks. But generalizing to real-world application needs to deal with multiple modalities.
>
> We thank the reviewer for raising this point. While multi-modality is indeed crucial for real-world social reasoning, the core contribution of Social-R1 is a modality-agnostic cognitive-reasoning framework based on SIP. Our method focuses on modelling the social-reasoning process itself, which is independent of whether cues are textual, visual, or auditory. Text was chosen as a controlled setting consistent with prior ToM and social-reasoning work.
> We fully agree that future social-reasoning agents should incorporate visual and audio cues, and we will extend Social-R1 toward multi-modal training in future work. This represents a promising but substantially different research direction beyond the scope of the current paper.
>
>
> > Weakness 3: Despite ToMBench-Hard being more challenging than existing benchmarks, it remains in a multiple-choice format. The multiple-choice nature is essentially a closed-ended evaluation, which may not fully capture the model's true social competence in open-ended, free-form interaction, or generative tasks. The model might be proficient at distinguishing between provided options but fail to exhibit original, appropriate social responses in broader social contexts.
>
> We sincerely thank the reviewer for raising this point. Our choice of a multiple-choice format is mainly motivated by three considerations:
>
> **(1) Controlled, unbiased and fully automated evaluation**
>
> Open-ended responses require LLM-as-a-Judge scoring in addition to costly human annotation, but recent studies have shown that LLM judges exhibit position bias [1], verbosity bias [2], and self-preference bias [3]. The multiple-choice setting allows us to eliminate these confounds and ensures that ToM difficulty arises purely from reasoning rather than stylistic or generative variability.
>
> **(2) Reliability and reproducibility**
>
> Open-ended scoring has high variance and is sensitive to phrasing and decoding randomness, which makes the evaluation less stable and less comparable across models [4]. In contrast, the multiple-choice format enables fully deterministic and reproducible evaluation—crucial for assessing models' social reasoning ability.
>
> **(3) Alignment with current research methodology**
>
> Most existing social-reasoning research (e.g., SocialIQa, Hi-ToM, ToMBench) also adopts a controlled, closed-ended design for the same reason: avoiding subjective evaluation noise.
> Future Direction
> We fully agree that open-ended, interactive evaluation is an important direction for assessing real-world social competence. In future work, we will extend ToMBench-Hard with a free-form generation track once a reliable, bias-controlled scoring protocol is available. We hope this clarifies the rationale behind our design.
>
> [1]Shi, Lin, Chiyu Ma, Wenhua Liang, Xingjian Diao, Weicheng Ma, and Soroush Vosoughi. "Judging the judges: A systematic study of position bias in llm-as-a-judge." arXiv preprint arXiv:2406.07791 (2024)
>
> [2] Saito, Keita, Akifumi Wachi, Koki Wataoka, and Youhei Akimoto. "Verbosity bias in preference labeling by large language models." arXiv preprint arXiv:2310.10076 (2023).
>
> [3]Wataoka, Koki, Tsubasa Takahashi, and Ryokan Ri. "Self-preference bias in llm-as-a-judge." arXiv preprint arXiv:2410.21819 (2024).
>
> [4]Schroeder, Kayla, and Zach Wood-Doughty. "Can you trust llm judgments? reliability of llm-as-a-judge." arXiv preprint arXiv:2412.12509 (2024).

---

> ### Author Response · Authors · 2025-11-25
> **Responses to Reviewer nPVH(2/2)**
>
> >Weakness4: Reinforcement Learning with Verified Rewards (RLVF) training, especially when combined with an additional reward model (the Social Thinking Reward Model), is typically more expensive and complex than traditional Supervised Learning (SL) and Instruction Tuning. This high computational cost may limit the application and reproducibility of the Social-R1 framework across the broader research community or for institutions with limited resources.
>
> Thank you for concern about the computational cost of RLVF training. We completely agree that RLVF is more resource-intensive than SL. We address this point from three perspectives:
>
> **(1) RLVF is a necessary and valuable methodological direction for enhancing social reasoning.**
>
> Prior work (e.g., DeepSeek-R1) has shown that RL is essential for improving structured reasoning. In our setting, verified rewards directly optimise explicit and interpretable reasoning trajectories rather than only surface-level outputs, leading to stronger social-reasoning performance—an ability that SL cannot directly provide.
>
> **(2)  Mitigating computational cost**
>
> Our pipeline is deliberately designed to be efficient. Social-R1 is trained with a small but strong reward-model backbone (Qwen3-4B) under the supervision of a compact dataset (socialthink-6k). This avoids more expensive training with larger reward models while achieving a good balance of cost, speed, and stability in multi-dimensional reward optimisation. Importantly, our framework enables small models (e.g., Social-R1-4B) to achieve social-reasoning performance comparable to much larger models (e.g., LLaMa3-70B), demonstrating that effective improvement does not require training large models.
>
> **(3) Ensuring reproducibility and accessibility**
>
> To enable broad adoption, we will release full reward-model weights, code, and training configurations, along with SFT-only and reward-filtered baselines for users who cannot run RL. These components ensure that the Social-R1 framework is reproducible without large-scale compute.

---

> > ### Author Response · Authors · 2025-12-03
> > **Thank you to  Reviewer nPVH**
> >
> > Thank you very much for your time and suggestions. We truly appreciate it.

---

### Official Review · Reviewer_T7Tx · 2025-11-04

**Soundness:** 3
**Presentation:** 3
**Contribution:** 3
**Rating:** 8
**Confidence:** 4

**Summary:**

This paper introduces Social-R1, a reinforcement learning framework for improving social intelligence in LLMs. The work makes three main contributions: ToMBench-Hard, a 900-question Theory of Mind benchmark spanning 6 dimensions; a social thinking reward model trained reasoning trajectories that are designed according to Social Information Processing (SIP) theory; and Social-R1, a reinforcement learning framework training framework that combines outcome-level and trajectory-level rewards using GRPO optimization. The authors find that the Social-R1 consistently outperforms strong reasoning LLMs across SocialIQA, SimpleToM, EmoBench, and MotiveBench and that this framework enables much smaller models to surpass or match larger models on these benchmarks. These results provide evidence that process-level thinking rewards should be used to supplement outcome-level rewards for human-like social intelligence in language models.

**Strengths:**

Overall, this paper is well-motivated and well-written, presenting a benchmark that is valuable for its comprehensiveness and complexity. ToMBench-Hard addresses real limitations in existing ToM evaluations through careful human curation across six ability dimensions (spanning emotion, desire, intention, knowledge, belief, and non-literal communication), achieving meaningful difficulty calibration where humans score 87% while state-of-the-art models struggle below 64%. The work makes an important empirical contribution by demonstrating that RLVF, previously confined to objective domains like math and coding, can effectively improve social reasoning and that process/reasoning-level rewards provide additional gains to solely outcome-based rewards. The results are compelling, particularly the finding that Social-R1-4B outperforms models with 10x more parameters across all benchmarks with consistent out-of-domain transfer.

**Weaknesses:**

The authors state that existing benchmarks have "exploitable patterns" and enable "superficial shortcuts" but don't provide much insight into what these are. Manipulations of perceptual access and asymmetric information may increase difficulty, but its not clear that this will prevent shortcut solutions or just allow models to index on different kinds of features (like transparency). Addressing this claim in a more systematic way would significantly strengthen the paper:
- Categorize specific exploitable patterns in existing benchmarks (Hi-ToM, ToMBench, etc.) and show that models exploit these patterns or provide citations to this effect (apologies if I missed this)
- Explain the design choice in ToMBench-Hard that addresses this and perform ablations showing that removing that design element reintroduces the shortcut
- Experiments verifying that the models aren't using new shortcuts.

Readers could benefit from more insight into the authors process in constructing reward model and RL optimization, as well as more analysis of the reward function itself:
- The reward design seems highly dependent on the authors choice of SIP and this choice isn't explained. Why not simulation theory (Goldman, 2006) or Theory-Theory (Gopnik & Wellman, 1994)? Explaining why SIP was chosen, and if possible, experiments comparing reward models informed by these other competing theories of ToM abilities in humans would strengthen this work.
- It seems like the outcome and format rewards plateau but the thinking reward does not -- can the authors provide any intuition for this? possible connection to the binarization of the thinking reward? Is there anything interpretable about the weighting coefficients that are learned that could explain the behavior of the reward function.

**Questions:**

In addition to addressing the points from weaknesses some minor points:
- There are a number of minor capitalization (line 156) and spelling issues, as well as missing/incorrect words especially in section 3.2 that made the text there a bit difficult to read.
- Section 3.3: Why was the reward model initialized from Qwen3-4B? Do the authors think the results would change with a different choice of reward model?
- I assume the sigmoid function only applied to the thinking reward in $R_i$ because the format and outcome rewards are already binary, but this could be made a bit more explicit.

---

> ### Author Response · Authors · 2025-11-25
> **Response to Reviewer T7Tx (1/2)**
>
> >Weakness1: The authors state that existing benchmarks have "exploitable patterns" and enable "superficial shortcuts" but don't provide much insight into what these are.
>
> We thank the reviewer for requesting a clearer explanation of “exploitable patterns.”
>
> **How ToMBench-Hard addresses these shortcuts**
>
> Prior work shows that existing ToM benchmarks [1] contain superficial cues that LLMs exploit instead of performing genuine belief reasoning. It demonstrates that LLM performance collapses under trivial wording changes—those capabilities vanish once tasks are slightly altered—indicating strong reliance on template-level shortcuts. [2] further identifies ten classes of surface heuristics (e.g., irrelevant distractors, unreliable testimony, sentiment-based guessing) that enable correct answers without mental-state tracking. ToMBench-Hard directly incorporates these manipulation strategies to break pattern-matching shortcuts and require genuine social reasoning for correct predictions.
>
> **Empirical verification**
>
> In the updated PDF (Appendix 6.1), we provide qualitative comparisons showing that even the strongest model (GPT-5) fails when perceptual-access cues are decoupled from the underlying social reasoning. These samples demonstrate that models cannot rely on surface features—such as transparency terms—to predict the correct answer, indicating that shortcut-based heuristics are effectively suppressed. Compared with the base model Qwen3-4B before training, our Social-R1-4B model shows clear improvements in interpreting social cues within scenarios and performing multi-step social reasoning.
>
> **Ablation Study**
>
> We conduct a 20-sample ablation study in which we added perceptual-access and asymmetric-information manipulations. Performance deteriorated substantially, with accuracy dropping from 100% to 40%. This reinstatement of shortcut-like behaviours indicates that these manipulations are crucial for preventing superficial pattern matching. We have uploaded two files (result w/wo shortcut.csv) to the anonymous GitHub linked on page 10 of the paper to show example cases and performance before and after the modifications.
>
> [1]Ullman, Tomer. "Large language models fail on trivial alterations to theory-of-mind tasks." arXiv preprint arXiv:2302.08399 (2023).
>
> [2] Nickel, Christian, Laura Schrewe, and Lucie Flek. "Probing the robustness of theory of mind in large language models." arXiv preprint arXiv:2410.06271 (2024).
>
> >Weakness 2:  The reward design seems highly dependent on the authors choice of SIP and this choice isn't explained. Why not simulation theory or Theory-Theory (Gopnik & Wellman, 1994)? Explaining why SIP was chosen.
>
> We sincerely thank the reviewer for raising this theoretical question. Below we clarify our choice of Social Information Processing (SIP) as the psychological backbone of the reward model and explain why Simulation Theory and Theory-Theory were not adopted as the primary framework.
>
> **(1) SIP provides an operational and step-wise structure that is highly suitable for practical reward modelling.**
>
> SIP explicitly decomposes social reasoning into empirically validated stages—(i) encoding social cues, (ii) interpreting and representing mental states, (iii) clarifying social goals, (iv) generating possible responses, (v) selecting a response, and (vi) evaluating the outcome. This step-wise and observable structure is essential for reward modelling because it enables us to define verifiable criteria for intermediate reasoning steps. In contrast, Simulation Theory emphasises internal perspective-taking, and Theory-Theory focuses on belief–desire–intention schemas, but neither offers a directly operationalisable pipeline for constructing reward signals.
>
> **(2) SIP reasoning stages closely align with the reasoning model’s “think-process”**
>
> As shown in Figure6 in Appendix 6, the model’s think-process when doing social reasoning tasks naturally mirrors the stages defined in SIP: the reasoning trajectory already involves perceiving cues, interpreting mental states, inferring goals, and generate responses, which makes SIP an promising structure for reward modelling.
>
> **(3) Our reward design implicitly incorporates key elements of both Theory-Theory and Simulation Theory theories.**
>
> Guided by SIP, the model performs (i) inference based on the specific agent’s cues (perspective-taking in Simulation Theory), and (ii) reasoning about how mental states lead to actions (Theory-Theory). Thus, choosing SIP does not exclude these theories; rather, SIP provides a unified and implementable abstraction that subsumes their useful components while remaining practical for reward modelling.
>
> >Weakness3:  There are a number of minor capitalization (line 156) and spelling issues, as well as missing/incorrect words especially in section 3.2.
>
> We thank the reviewer for pointing out these writing issues. We have carefully revised the entire manuscript. The revised PDF has been updated accordingly.

---

> ### Author Response · Authors · 2025-11-25
> **Response to Reviewer T7Tx (2/2)**
>
> >It seems like the outcome and format rewards plateau but the thinking reward does not -- can the authors provide any intuition for this? possible connection to the binarization of the thinking reward? Is there anything interpretable about the weighting coefficients that are learned that could explain the behavior of the reward function.
>
> Thank you for this insightful question. We agree that Figure 2(c) shows early plateauing for outcome and format rewards, whereas the thinking reward continues to improve. This behaviour is expected given the fundamental differences across the three reward signals.
>
> **(1) Why outcome and format plateau**
>
> Outcome and format rewards are binary (0/1) and are already close to saturation in the base model. Under GRPO, these rewards quickly reach ceiling performance, and once most trajectories achieve the correct answer and valid format, there is little remaining optimisation signal. This naturally leads to flat curves.
>
> **(2) Why the thinking reward does not plateau**
>
> In contrast, the thinking reward evaluates multi-step reasoning quality and provides a fine-grained supervision signal. The base model is far from saturation in this dimension. Therefore, GRPO continues to exploit this headroom, leading to gradual improvement rather than an early plateau.
>
> **(3) Connection to binarisation/sigmoid**
>
> The non-plateau of the thinking reward is not caused by binarisation. Only the outcome and format rewards are binary. The thinking reward is continuous, and the sigmoid merely rescales it into (0,1) for optimisation stability—it does not discretise or compress away meaningful differences. Thus, the thinking reward continues to provide informative gradients even when the other signals saturate.
>
> **(4) Weighting coefficients**
>
> To clarify, the weighting coefficients of the three rewards are not learned in our framework; they are all fixed to 1. Thus, the observed behaviour does not stem from weight adaptation but from the inherent properties of the reward components themselves.  We have clarified these points in the revised manuscript .
>
> >Why was the reward model initialized from Qwen3-4B? Do the authors think the results would change with a different choice of reward model?
>
> We thank the reviewer for the thoughtful question about our choice of Qwen3-4B as the reward-model backbone. We chose Qwen3-4B for three reasons:
>
> **(1) Smaller reward-model backbones are widely adopted and offer better stability–efficiency trade-offs [1]**
>
> From prior work, small reward models strike a good balance between cost, speed, and stability in multi-dimensional reward optimisation. RLHF pipelines show that small backbones are sufficient and often preferred for stability and efficiency.
>
> **(2) Qwen3-4B offers strong reasoning performance among small models, improving RM stability.**
>
> A recent systematic study [2] shows that Qwen variants outperform other small LLMs (LLaMA, Mistral, Gemma) on multiple reasoning and judging benchmarks. As a strong reasoning model [3], Qwen3-4B serves as a stable base for a reward model whose role is to evaluate social-reasoning quality rather than to generate complex reasoning chains. Thus, Qwen3-4B provides an efficient yet sufficiently capable backbone for modelling SIP-based social-reasoning rewards.
>
> **(3) Using a small backbone reduces confounding effects introduced by latent knowledge in larger models**
>
> Larger reward models (e.g., >8B) risk over-relying on their own world knowledge or stylistic priors when evaluating trajectories(from 4B) . A smaller backbone reduces such confounds and ensures that the reward function reflects our SIP-based criteria rather than implicit memorisation, improving interpretability of the reward.
>
> We expect larger reward models to yield quantitative differences without altering the qualitative conclusions. Running these experiments requires substantial compute, so they were not included in the initial submission. We are running additional experiments with a Qwen3-8B reward model. Due to limited computational resources, the final results require more time. We are actively running the experiment and will update the manuscript as soon as they are completed. We guarantee that the full results will be included in the final version.
>
>
> [1] Ouyang, Long, Jeffrey Wu, Xu Jiang et al. "Training language models to follow instructions with human feedback." NeurIPS 2022
>
> [2] Kian Ahrabian et al.  A Systematic Analysis of Base Model Choice for Reward Modeling. EMNLP 2025
>
> [3] Jayarao, Pratik, et al. "Explicit Reasoning Makes Better Judges: A Systematic Study on Accuracy, Efficiency, and Robustness." arXiv 2025
>
> >I assume the sigmoid function only applied to the thinking reward in because the format and outcome rewards are already binary, but this could be made a bit more explicit.
>
> Yes, the sigmoid is applied only to the thinking reward. It maps the thinking reward to a continuous score in (0, 1), and we have clarified this in the revised manuscript.

---

> ### Author Response · Authors · 2025-12-03
> **Thank you to  Reviewer T7Tx**
>
> Thank you very much for your time and for recognising our work. We sincerely appreciate your constructive suggestions, which have helped us further clarify and strengthen the paper.

---

### Author Response · Authors · 2025-12-03
**Rebuttal Clarifications to the Area Chair(1/2)**

Dear Area Chair,
Thank you very much for taking the additional time to review our paper. For your convenience, we provide a concise summary of our work and the key recognitions from all reviewers.

**Work Summary**

Our paper makes three main contributions:
**(1)** ToMBench-Hard, a human-curated benchmark designed to evaluate genuine Theory-of-Mind reasoning by removing shortcut patterns through principled psychological manipulations. **(2)** A method extending RLVF to social reasoning using a trajectory-level thinking reward grounded in Social Information Processing theory. **(3)** Empirical improvements in social reasoning, where a 4B model trained with outcome and thinking rewards achieves strong in-domain and out-of-domain performance, surpassing substantially 70B model.

**Recognition from All Reviewers**

All four reviewers provide consistent recognition of the significance, methodological soundness, and empirical impact of our work.

**1. Community Contribution and Importance of the Problem**
- R3 (rxsk) states that the paper addresses a significant and timely problem—improving LLM social reasoning—and that ToMBench-Hard is a valuable community resource for evaluating true social inference.

**2. Comprehensive and Well-Curated Benchmark**
- R1 (T7Tx) praises ToMBench-Hard as a valuable, comprehensive, and carefully curated benchmark covering six ToM dimensions, with a meaningful human–model gap (87% vs. <64%). R2 (nPVH) highlights that the benchmark eliminates shortcut exploitation and authentically evaluates complex ToM scenarios such as deception and belief revision. R4 (k6s4) notes that it provides a harder, multi-party, context-dependent ToM evaluation with a clear human–model gap.

**3. Novel and Well-Grounded Methodology**
- R1 (T7Tx) recognises the importance of showing that RLVF—traditionally used in math and coding—effectively improves social reasoning and that process-level rewards add value. R2 (nPVH) states that combining outcome-based rewards with trajectory-based thinking rewards is the correct direction for enhancing LLM social intelligence. R3 (rxsk) praises the originality of integrating process-level supervision with outcome rewards and highlights the strong grounding in SIP theory. R4 (k6s4) agrees that the motivation for combining process- and outcome-level supervision is clear.

**4. Strong Empirical Contribution**
- R1 (T7Tx) emphasises the compelling empirical findings, particularly that Social-R1-4B outperforms models with 10× more parameters and shows strong out-of-domain transfer. R2 (nPVH) highlights that Social-R1-4B surpasses LLaMA3-70B, validating the effectiveness of combining outcome and thinking rewards. R3 (rxsk) views the empirical evaluation as strong evidence of improved social reasoning.

---

> ### Author Response · Authors · 2025-12-03
> **Rebuttal Clarifications to the Area Chair(2/2)**
>
> **Rebuttal Updates**
>
> In the rebuttal, we directly answer the main concerns raised by all four reviewers and add new analyses, experiments, and materials to the appendix and an anonymous GitHub repository (link on page 10):
> - R1 (T7Tx).
>
> We clarify what “exploitable patterns” mean in existing ToM benchmarks, link them to prior findings, and show how ToMBench-Hard removes these shortcuts. We add qualitative examples and a 20-sample ablation study (Appendix 6 & GitHub) demonstrating that our manipulations substantially reduce shortcut-like exploitability. We further explain why we choose SIP as the grounding theory for the thinking reward, clarify why outcome/format rewards plateau while the thinking reward continues to improve, and make the use of the sigmoid and fixed reward weights explicit. We also justify the choice of Qwen3-4B as the reward-model backbone.
>
> - R2 (nPVH).
>
> We clarify that Social-R1 is a modality-agnostic framework, and that we use text as a controlled setting consistent with prior ToM work. We clarify the multiple-choice design as a reliable, unbiased, and reproducible evaluation protocol compared with open-ended social tasks that rely on LLM-as-judge and suffer from strong judge biases. We also address computational cost, explaining how our design (small RM backbone, compact SocialThink-6k dataset, 4B base model) keeps RLVF training efficient, and we commit to releasing reward-model weights, scripts, and data to support reproducibility.
>
> - R3 (rxsk)
>
>  We strengthen dataset transparency by adding detailed distributions, annotation protocols, and representative examples in the appendix, and by releasing an anonymised subset of ToMBench-Hard in an anonymous repository for direct inspection. To address the circularity concern, we clarify that GPT-5 only provides coarse-grained scores conditioned on gold answers and never supplies gold trajectories or templates. We further add cross-model scoring analyses showing that the ou Thinking Reward Model aligns with human judgements and generalises across models rather than mimicking GPT-5’s style. We also revise the comparison with LLaMA3-70B to clearly frame it as evidence that process-level RL on high-quality ToM data unlocks social reasoning that does not emerge from scale alone.
>
> - R4 (k6s4)
>
> We clarify that ToMBench-Hard intentionally emphasises challenging, under-tested ToM sub-skills rather than enforcing a uniform distribution. We explain that ground truth arises from multi-annotator consensus and that human accuracy below 1.0—also common in prior large-scale benchmarks—reflects natural human variability rather than unreliable labels. We add further tables and discussion to show that the reported “anomalies” reflect model limitations and differences in coverage and difficulty, rather than flaws in the construct validity of ToMBench-Hard.
>
>
> **Once again, we sincerely appreciate your time and consideration.**

---

### Meta-Review · Area_Chair_Neux · 2026-01-06

**Summary:**

This paper introduces Social-R1, a RL framework designed to enhance social intelligence in large language models by combining outcome-based rewards with trajectory-level thinking rewards. The authors also present ToMBench-Hard, a human-curated multiple-choice benchmark aimed at evaluating Theory-of-Mind (ToM) capabilities while mitigating shortcut exploitation.

**Reviewer Concerns:**

The reviewers acknowledge the significance of the research problem and the novelty of integrating Social Information Processing (SIP) theory into reward modeling. Several concerns were successfully addressed by the rebuttal, including the release of a dataset subset for further review, clarification for the SIP framework and the efficiency of the RLVR pipeline. However, several major concerns raised by the reviewers remain outstanding. 1) As noted by Reviewer k6s4, the "anomalies" in Table 1 where explicit reasoning (CoT) actually degrades performance for several base models raise concerns about the benchmark's reliability. 2) The reliance on o3 and GPT-5 to generate gold trajectories and assign quality scores may result in overfitting of the stylistic preferences and internal biases rather than generalized social intelligence. 3) The ToMBench-Hard distribution is skewed toward multi-choice format and specific categories like "Intention", which weakens the evaluation.

**Reviewer Scores:**

Reviewer T7Tx is likely remains at a score of 8. Reviewer nPVH's concerns is relatively minor and less informative, and thus the rating is downweighted. Reviewer rxsk and k6s4 may remains 2 due to the remaining concerns.

---

### Decision · Program_Chairs · 2026-01-26

Reject